# Deep Mean Field Theory: Layerwise Variance and Width Variation as Methods to Control Gradient Explosion

## Abstract

A recent line of work has studied the statistical properties of neural networks to great success from a *mean field theory* perspective, making and verifying very precise predictions of neural network behavior and test time performance. In this paper, we build upon these works to explore two methods for taming the behaviors of random residual networks (with only fully connected layers and no batchnorm). The first method is *width variation (WV)*, i.e. varying the widths of layers as a function of depth. We show that width decay reduces gradient explosion without affecting the mean forward dynamics of the random network. The second method is *variance variation (VV)*, i.e. changing the initialization variances of weights and biases over depth. We show VV, used appropriately, can reduce gradient explosion of tanh and ReLU resnets from $\exp(\Theta(\sqrt{L}))$ and $\exp(\Theta(L))$ respectively to constant $\Theta(1)$. A complete phase-diagram is derived for how variance decay affects different dynamics, such as those of gradient and activation norms. In particular, we show the existence of many phase transitions where these dynamics switch between exponential, polynomial, logarithmic, and even constant behaviors. Using the obtained mean field theory, we are able to track surprisingly well how VV at initialization time affects training and test time performance on MNIST after a set number of epochs: the level sets of test/train set accuracies coincide with the level sets of the expectations of certain gradient norms or of metric expressivity (as defined in Yang and Schoenholz (2017)), a measure of expansion in a random neural network. Based on insights from past works in deep mean field theory and information geometry, we also provide a new perspective on the gradient explosion/vanishing problems: they lead to ill-conditioning of the Fisher information matrix, causing optimization troubles.

## 1 Introduction

*Deep mean field theory* studies how random neural networks behave with increasing depth, as the width goes to infinity. In this limit, several pieces of seminal work used statistical physics (Derrida and Pomeau, 1986; Sompolinsky et al., 1988) and Gaussian Processes (Neal, 2012) to show that neural networks exhibit remarkable regularity. Mean field theory also has a substantial history studying Boltzmann machines (Ackley et al., 1985) and sigmoid belief networks (Saul et al., 1996).

Recently, a number of results have revitalized the use of mean field theory in deep learning, with a focus on addressing practical design questions. In Poole et al. (2016), mean field theory is combined with Riemannian geometry to quantify the expressivity of random neural networks. In Schoenholz et al. (2017) and Yang and Schoenholz (2017), a study of the critical phenomena of mean field neural networks and residual networks[1] is leveraged to theoretically predict test time relative performance of differential initialization schemes. Additionally, Choromanska et al. (2015) and Pennington and Bahri (2017) have used related techniques to investigate properties of the loss landscape of deep networks. Together these results have helped a large number of experimental observations onto more rigorous footing (Montfar et al., 2014; Glorot and Bengio, 2010; Bertschinger et al., 2005). Finally, deep mean field theory has proven to be a necessary underpinning for studies using random matrix theory to

---

[1]without batchnorm and with only fully connected layers

understand dynamical isometry in random neural networks (Pennington et al., 2017; Pennington and Worah, 2017). Overall, a program is emerging toward building a mean field theory for state-of-the-art neural architectures as used in the wild, so as to provide optimal initialization parameters quickly for any deep learning practitioner.

In this paper, we contribute to this program by studying how width variation (WV), as practiced commonly, can change the behavior of quantities mentioned above, with gradient norm being of central concern. We find that WV can dramatically reduce gradient explosion without affecting the mean dynamics of forward computation, such as the activation norms, although possibly increasing deviation from the mean in the process (Section 6).

We also study a second method, variance variation (VV), for manipulating the mean field dynamics of a random neural network (Section 7 and Appendix B). In this paper, we focus on its application to tanh and ReLU residual networks, where we show that VV can dramatically ameliorate gradient explosion, and in the case of ReLU resnet, activation explosion [2]. Affirming the results of Yang and Schoenholz (2017) and predicted by our theory, VV improves performances of tanh and ReLU resnets through these means.

Previous works (Poole et al., 2016; Schoenholz et al., 2017; Yang and Schoenholz, 2017) have focused on how network architecture and activation functions affect the dynamics of mean field quantities, subject to the constraint that initialization variances and widths are constant across layers. In each combination of (architecture, activation), the mean field dynamics have the same kinds of asymptotics regardless of the variances. For example, tanh feedforward networks have $\exp(\Theta(l))$ forward and backward dynamics, while tanh residual networks have $\text{poly}(l)$ forward and $\exp(\Theta(\sqrt{l}))$ backward dynamics. Such asymptotics were considered characteristics of the (architecture, activation) combination (Yang and Schoenholz, 2017). We show by counterexample that *this perception is erroneous*. In fact, as discussed above, WV can control the gradient dynamics arbitrarily and VV can control forward and backward dynamics jointly, all without changing the network architecture or activation. To the best of our knowledge, this is the first time methods for reducing gradient explosion or vanishing have been proposed that vary initialization variance and/or width across layers.

With regard to ReLU resnets, we find that gradient norms and "metric expressivity" (as introduced in Yang and Schoenholz (2017), also defined in Defn 4.2), make surprisingly good predictors, respectively in two separate phases, of how VV at initialization affects performance after a fixed amount of training time (Section 7.1). However, in one of these phases, *larger* gradient explosion seems to cause *better* performance, with no alternative course of explanation. In this paper we have no answer for why this occurs but hope to elucidate it for future work. With regard to tanh resnets, we find that, just as in Yang and Schoenholz (2017), the optimal initialization balances trainability and expressivity: Decaying the variance too little means we suffer from gradient explosion, but decaying the variance too much means we suffer from not enough metric expressivity.

We want to stress that in this work, by "performance" we do not mean absolute performance but rather *relative performance* between different initialization schemes. For example, we do not claim to know what initialization scheme is needed to make a particular neural network architecture solve ImageNet, but rather, conditioned on the architecture, whether one initialization is better than another in terms of test set accuracy after the same amount of training iterations.

Before we begin the mean field analysis, we present a perspective on gradient explosion/vanishing problem from a combination of mean field theory and information geometry, which posits that such problem manifests in the ill-conditioning of the Fisher information matrix.

## 2 GRADIENT EXPLOSION/VANISHING: AN INFORMATION GEOMETRIC PERSPECTIVE

Given a parametric family of probability distributions on $\mathbb{R}^m$, $P := \{P_\theta\}_\theta$ with $\theta = (\theta_1, \dots, \theta_n)$ in $\mathbb{R}^n$, its Fisher information matrix is defined as $F(\theta) := [\mathbb{E}_{z \sim P_\theta}(\partial_i \log P_\theta(z))(\partial_j \log P_\theta(z))]_{i,j=1}^n$ (here $\partial_i$ is partial derivative against $\theta_i$). It is known from information geometry that, under regularity conditions, $P$ forms a Riemannian manifold with $\theta \mapsto P_\theta$ as its coordinate map and $F(\theta)$ as

---

[2]This is the phenomenon where in deep ReLU resnet, the activation vectors blow up exponentially with depth, invalidating the computation with `infs`; see Yang and Schoenholz (2017)

its Riemannian metric tensor (*a fortiori* it is positive definite) (Amari, 2016). This fact is most famously used in the natural gradient method, which, akin to second order methods, computes from a gradient vector $\partial E/\partial \theta$ a "natural direction of greatest descent" $F(\theta)^{-1}\partial E/\partial \theta$ that is invariant to reparametrization $\theta \to \theta'$ (Amari, 1998). This method and related ideas have been applied to great success in supervised, unsupervised, and reinforcement learning (for example, Pascanu and Bengio (2013); Desjardins et al. (2015); Martens and Grosse (2015); Grosse and Martens (2016); Wu et al. (2017)). An $F(\theta)$ with eigenvalues all approximately equal means that the neighborhood around $P_\theta$ is isotropically curved and the gradient is approximately just the natural gradient up to a multiplicative constant. Conversely, an $F(\theta)$ with a large condition number $\kappa(F(\theta))$ (the ratio of the largest over the smallest eigenvalue) means that the gradient is a poor proxy for the natural gradient and thus is much less efficient. From another angle, $F(\theta)$ is also the Hessian of the KL divergence $\tau \mapsto \mathrm{KL}(P_\theta \| P_\tau)$ at $\tau = \theta$. If we were simply to minimize this KL divergence through gradient descent, then the number of iterations to convergence is proportional to $\kappa(F(\theta))$ (in general, there is a lower bound of $\Omega(\sqrt{\kappa(F(\theta))})$ for first order methods satisfying a mild condition) (Nesterov, 2004).

For a random deep network (residual or not) suffering from gradient explosion in the mean, we show heuristically in this section that the condition number of its Fisher information matrix is exponentially large in depth with high probability [3]. First partition $\theta$ into groups of parameters according to layer, $\theta = (\theta_{11}, \theta_{12}, \ldots, \theta_{1k_1}, \theta_{21}, \ldots, \theta_{2k_2}, \ldots, \theta_{L1}, \ldots, \theta_{Lk_L})$, with $\theta_{lj}$ denoting parameter $j$ of layer $l$. We can then partition the Fisher information matrix $F(\theta)$ into blocks, with the diagonal blocks having sizes $k_1 \times k_1, k_2 \times k_2, \ldots, k_L \times k_L$.

According to the Hermitian min-max theorem, the largest eigenvalue of $F(\theta)$ is given by $\max_{\|x\|=1} x^T F(\theta)x$ and the smallest eigenvalue is given by $\min_{\|x\|=1} x^T F(\theta)x$ (both are positive as $F(\theta)$ is positive definite under our regularity assumptions). Thus $\kappa(F(\theta))$ equals to their ratio and is lower bounded by a ratio of extremal diagonal terms $\max_{lj} F(\theta)_{lj,lj}/\min_{lj} F(\theta)_{lj,lj}$. Let $\langle Y(\theta) \rangle$ be the expectation of $Y(\theta)$ with respect to random initialization of $\theta$ in some fixed method. Suppose there is gradient explosion such that $\langle \mathrm{E}_z(\partial_{lj}\log P_\theta(z))^2 \rangle \in [\exp(c'l), \exp(C'l)]$ for universal constants $c', C' > 0$ independent of $j$ (this is true, for example, for feedforward tanh networks initialized in the chaotic region Schoenholz et al. (2017)). By concentration of measure phenomenon (as seen in Daniely et al. (2016); Poole et al. (2016); Schoenholz et al. (2017); Yang and Schoenholz (2017) and this work), over randomization of parameters, $\mathrm{E}_z(\partial_{lj}\log P_\theta(z))^2$ will in fact concentrate around its mean as width goes to infinity. Thus we have, with high probability, that diagonal entries $F(\theta)_{lj,lj} = \mathrm{E}_z(\partial_{lj}\log P_\theta(z))^2 \in [\exp(cl), \exp(Cl)]$ for some new constants $0 < c < c' < C' < C$. Then the ratio $\max_{lj} F(\theta)_{lj,lj}/\min_{lj} F(\theta)_{lj,lj}$ is at least $\exp(cL)/\exp(C1) = \exp(cL - C)$, so that the $\kappa(F(\theta))$ is exponential in $L$. The argument can be easily modified to accommodate gradient vanishing and other rates of gradient explosion/vanishing like $\exp(\Theta(l^\alpha))$.

Thus such gradient dynamical problems cause the gradient to deviate from the natural gradient exponentially in an appropriate sense and violate the information geometry of the information manifold $P_\theta$. For the case of minimizing KL divergence from a specific distribution, they directly cause the number of gradient descent iterations to diverge exponentially. These issues cannot be solved by just adjusting the learning rate (though it can somewhat ameliorate the problem by taking conservative steps in such ill-conditioned regions).

## 3 BACKGROUND

The desire to understand how initialization can affect final performances of a deep neural network has led to a resurgence of mean field techniques, this time applied to deep learning. A series of papers (Poole et al., 2016; Schoenholz et al., 2017; Yang and Schoenholz, 2017; Pennington et al., 2017) have established the depth-wise dynamics of random neural networks (i.e. networks at initialization time). For example, Poole et al. (2016) showed that, in a random tanh classical feedforward neural network, activation norm converges exponentially fast in depth to a constant value, and so does the angle between images of two different input vectors at successive depths, which the authors proposed as a measure of expressivity that Yang and Schoenholz (2017) called "angular expressivity." Schoenholz et al. (2017) then showed that the gradient norm of such a random network suffers from exponential explosion or vanishing during the course of backpropagation. But when the initialization variances

---

[3]This can be made rigorous in a straightforward, though somewhat tedious, way, but we will not do so here

lie on a "critical curve," the gradient is neither vanishing nor exploding, and, more importantly, the networks initialized on this "critical line" has the best *test time performance* after training for a fixed number of iterations. The mean field framework was extended to residual networks (with only fully connected layers and no batchnorm) in Yang and Schoenholz (2017). There the authors showed that just by adding a skip connection to the feedforward network, the dynamics of a tanh network becomes subexponential. More crucially, they investigated both tanh and ReLU residual networks, and found that whereas gradient dynamics controls the test time performances of tanh resnets, "expressivity" controls those of ReLU resnets. This expressivity is, roughly speaking, how much distance a random network on average puts between two different input vectors; it was aptly named "metric expressivity." On the other hand, the "angular expressivity" proposed in Poole et al. (2016) (how much angle the network puts between two input vectors, as explained above) was not found to be predictive of relative test time performance of either tanh or ReLU resnets.

More precisely, the optimal initialization scheme for tanh resnet seems to strike a delicate balance between trainability and expressivity, in that weight variance too large causes too much gradient explosion and causes training to fail, whereas weight variance too small causes the typical network to collapse to a constant function Yang and Schoenholz (2017). The optimal variance $\sigma_w^2$ satisfies $\sigma_w^2 L = \text{const}$ where $L$ is depth. On the other hand, ReLU resnets have completely different behavior with respect to initialization variance; here the best initialization scheme is obtained by maximizing the weight variance (and as a consequence also maximizing the metric expressivity) without overflowing activation values of deeper layers into numerical `inf`s. Indeed, trainability seems to not be a problem at all, as the gradient norm of weight parameters at each layer stays constant within $O(1)$ over the course of backpropagation.

In this paper, we extend the results of Yang and Schoenholz (2017) to include depthwise variation of widths and of variances. We show that they can be used to great effect to reduce gradient explosion as well as manipulating the expressivity (metric or angular) of the random network. Corroborating Yang and Schoenholz (2017), we find that they improve tanh resnet performance by taming gradient dynamics and improve ReLU resnet performance by preventing activations from numerically overflowing while maximizing metric expressivity. However, in certain regimes, worsening gradient explosion can mysteriously make ReLU resnet perform *better*, and we currently do not know how to explain this phenomenon.

## 4 PRELIMINARIES

**Notations and Settings.** We adopt the notations of Yang and Schoenholz (2017) and review them briefly. Consider a vanilla feedforward neural network of $L$ layers, with each layer $l$ having $N^{(l)}$ neurons; here layer 0 is the input layer. Let $x^{(0)} = (x_1^{(0)}, \ldots, x_{N^{(0)}}^{(0)})$ be the input vector to the network, and let $x^{(l)}$ for $l > 0$ be the activation of layer $l$. Then a neural network is given by the equations $x_i^{(l)} = \phi(h_i^{(l)}), h_i^{(l)} = \sum_{j=1}^{N^{(l-1)}} w_{ij}^{(l)} x_j^{(l-1)} + b_i^{(l)}$ where (i) $h^{(l)}$ is the pre-activation at layer $l$, (ii) $w^{(l)}$ is the weight matrix, (iii) $b^{(l)}$ is the bias vector, and (iv) $\phi$ is a nonlinearity, for example tanh or ReLU, which is applied coordinatewise to its input. To lighten up notation, we suppress the explicit layer numbers $l$ and write $x_i = \phi(h_i), h_i = \sum_{j=1}^{N} w_{ij}\underline{x}_j + b_i$ where $\bullet$ implicitly denotes $\bullet^{(l)}$, and $\underline{\bullet}$ denotes $\bullet^{(l-1)}$ (and analogously, $\overline{\bullet}$ denotes $\bullet^{(l+1)}$). When the widths are constant $N^{(l)} = N^{(m)}, \forall m, l$, residual network (He et al., 2016a;b) adds an *identity connection* or *skip shortcut* that "jumps ahead" every couple layers. We adopt one of the simplified residual architectures defined in Yang and Schoenholz (2017) for ease of analysis [4], in which every *residual block* is given by

$$x_i = \sum_{j=1}^{M} v_{ij}\phi(h_j) + a_i + \underline{x}_i, \qquad\qquad h_i = \sum_{j=1}^{N} w_{ij}\underline{x}_j + b_i.$$

where $M^{(l)}$ is the width of the "hidden layer" of the residual block, $(v_{ij}^{(l)})_{i,j=1}^{N^{(l)},M^{(l)}}$ is a new set of weights and $(a_i^{(l)})_{i=1}^{N^{(l)}}$ is a new set of biases for every layer $l$. If we were to change the width of a

---

[4]It is called *full residual network* by Yang and Schoenholz (2017), but in this paper, we will simply assume this architecture whenever we say *residual network*.

residual network, as is done in practice, we need to insert *"projection" residual blocks* (He et al., 2016a;b) every couple layers. We assume the following simplified projection residual block in this paper, for the ease of presentation [5]:

$$x_i = \sum_{j=1}^{M} v_{ij}\phi(h_j) + y_i + a_i, \qquad h_i = \sum_{j=1}^{N} w_{ij}\underline{x}_j + b_i, \qquad y_i = \sum_{j=1}^{N} \pi_{ij}\underline{x}_j.$$

where $y = (y_i)_{i=1}^{N}$ is a "projection" of $x = (x_i)_{i=1}^{N}$ [6], and $(\pi_{ij})_{i,j=1}^{N,N}$ is the "projection" matrix. Note that we only consider fully-connected affine layers instead of convolutional layers.

Deep mean field theory is interested in the "average behavior" of these network when the weights and biases, $w_{ij}^{(l)}, b_i^{(l)}, \pi_{ij}^{(l)}, v_{ij}^{(l)}$, and $a_i^{(l)}$ are sampled i.i.d. from Gaussian distributions. Following standard practice, they respectively have standard deviations $\sigma_w^{(l)}/\sqrt{N^{(l-1)}}, \sigma_b^{(l)}, \sigma_\pi^{(l)}/\sqrt{N^{(l-1)}}, \sigma_v^{(l)}/\sqrt{M^{(l)}}$, and $\sigma_a^{(l)}$; here we normalize the variance of weight parameters so that, for example, the variance of each $h_i$ is $\sigma_w^2$, assuming each $\underline{x}_j$ is fixed. While previous works have all focused on fixing $\sigma_\bullet^{(l)}$ to be constant across depth, in this paper we are interested in studying varying $\sigma_\bullet^{(l)}$. In particular, other than $\sigma_\pi^{(l)}$ which we fix at 1 across depth (so that "projection" doesn't act like an "expansion" or "contraction"), we let $\sigma_\bullet^{(l)} = \sigma_\bullet l^{-\beta_\bullet}$ for each $\bullet = v, w, a, b$, where $\sigma_\bullet, \beta_\bullet$ are constant w.r.t. $l$. Hereafter the bar notation $\underline{\bullet}, \bullet, \overline{\bullet}$ do not apply to $\sigma$s, so that, by $\sigma_a$, for example, we always mean the constant $\sigma_a$.

We make the same statistical assumptions as in Yang and Schoenholz (2017). In the interest of space, we relegate their discussion to Appendix A.

**Mean Field Quantities.**  Now we define the central quantities studied in this paper. Inevitably, as deep mean field theory analyzes neural networks closer and closer to those used in practice, the variables and notations become more and more complex; our paper is no different. We have however included a glossary of symbols (Table A.1) that will hopefully reduce notation confusions to the first time reader.

**Definition 4.1.** Fix an input $x^{(0)}$. Define the **length quantities** $\mathbf{q}^{(l)} := \langle \|h^{(l)}\|^2 \rangle / M^{(l)} = \langle (h_1^{(l)})^2 \rangle$ and $\mathbf{p}^{(l)} := \langle \|x^{(l)}\|^2 \rangle / N^{(l)} = \langle (x_1^{(l)})^2 \rangle$ for $l > 0$ and $\mathbf{p}^{(0)} = \|x^{(0)}\|^2/N$. Here (and in the following definitions) the expectations $\langle \bullet \rangle$ are taken over all random initialization of weights and biases for all layers $l$, as $N^{(l)}, M^{(l)} \to \infty$ (large width limit). Note that in our definition, the index 1 can be replaced by any other index by Axiom 1. Thus $\mathbf{p}^{(l)}$ is the typical magnitude (squared) of a neuronal activation at layer $l$.

**Definition 4.2.** Fix two inputs $x^{(0)}$ and $x^{(0)\prime}$. We write $\bullet\prime$ to denote a quantity $\bullet$ with respect to the input $x^{(0)\prime}$. Then define **the correlation quantities** $\boldsymbol{\gamma}^{(l)} := \langle h^{(l)} \cdot h^{(l)\prime} \rangle / M^{(l)} = \langle h_1^{(l)} h_1^{(l)\prime} \rangle$ and $\boldsymbol{\lambda}^{(l)} := \langle x^{(l)} \cdot x^{(l)\prime} \rangle / N^{(l)} = \langle x_1^{(l)} x_1^{(l)\prime} \rangle$ for $l > 0$ and $\boldsymbol{\lambda}^{(0)} = x^{(0)} \cdot x^{(0)\prime}/N^{(0)}$. Again, here the index 1 does not matter by Axiom 1. By **metric expressivity**, we mean $\mathbf{s}^{(l)} := \frac{1}{2N}\langle \|x^{(l)} - x^{(l)\prime}\|^2 \rangle = \frac{1}{2N}(\langle \|x^{(l)}\|^2 \rangle + \langle \|x^{(l)\prime}\|^2 \rangle - 2\langle x^{(l)} \cdot x^{(l)\prime} \rangle) = \frac{1}{2}(\mathbf{p}^{(l)} + \mathbf{p}^{(l)\prime}) - \boldsymbol{\gamma}^{(l)}$. Additionally, define **the cosine distance quantities** $\mathbf{e}^{(l)} := \boldsymbol{\gamma}^{(l)}/\sqrt{\mathbf{p}^{(l)}\mathbf{p}^{(l)\prime}}$ and $\mathbf{c}^{(l)} := \boldsymbol{\lambda}^{(l)}/\sqrt{\mathbf{q}^{(l)}\mathbf{q}^{(l)\prime}}$, and we will also call $\mathbf{e}^{(l)}$ **angular expressivity**.

Following Yang and Schoenholz (2017), we assume $\mathbf{p}^{(0)} = \mathbf{p}^{(0)\prime}$ for the ease of presentation, but this is a nonessential assumption. Then, as we will see, $\mathbf{p}^{(l)} = \mathbf{p}^{(l)\prime}, \mathbf{q}^{(l)} = \mathbf{q}^{(l)\prime}$ for all $l$, and as a result, $\mathbf{e}^{(l)} = \boldsymbol{\gamma}^{(l)}/\mathbf{p}^{(l)}$ and $\mathbf{s}^{(l)} = \mathbf{p}^{(l)} - \boldsymbol{\gamma}^{(l)} = (1 - \mathbf{e}^{(l)})\mathbf{p}^{(l)}$.

**Definition 4.3.** Fix an input $x^{(0)}$ and a gradient vector $(\partial E/\partial x_i^{(L)})_i$ of some loss function $E$ with respect to the last layer $x^{(L)}$. Then define **the gradient quantities** $\boldsymbol{\chi}^{(l)} := \langle (\partial E/\partial x_1^{(l)})^2 \rangle, \boldsymbol{\chi}_\bullet^{(l)} := \langle (\partial E/\partial \bullet_1^{(l)})^2 \rangle$ for $\bullet = a, b$, and $\boldsymbol{\chi}_\bullet^{(l)} := \langle (\partial E/\partial \bullet_{11}^{(l)})^2 \rangle$ for $\bullet = w, v$. Here the expectations are

---

[5]If the reader is disappointed by our simplifications, he or she is encouraged to consult He et al. (2016b), which empirically explored many variations of residual blocks.

[6]"projection" is in quotes because the dimension $N^{(l)}$ can be less than or greater than $N^{(l-1)}$; but we will follow the traditional wording.

taken with Axiom 2 in mind, over both random initialization of forward and backward weights and biases, as $N \to \infty$ (large width limit). Again, the index 1 or 11 does not matter by Axiom 1.

## 5 OVERVIEW OF RESULTS AND TECHNIQUES

Just as in previous works in deep mean field theory (Poole et al., 2016; Schoenholz et al., 2017; Yang and Schoenholz, 2017), the primary tool for investigating behaviors of large width networks is the central limit theorem. Every time the activations of the previous layer pass through an affine layer whose weights are sampled i.i.d., the output is a sum of a large number of random variables, and thus follows an approximately Gaussian law. The output of the next nonlinearity is then a nonlinear transform of a Gaussian variable, with computable mean and variance. Repeating this logic gives us a depthwise dynamical system of the activation random variables. The gradient dynamics can be similarly derived, assuming Axiom 2. Table A.2 summarizes the equations governing the dynamics of various mean field quantities when we vary variances and/or widths.

We are interested in the asymptotics of such dynamics, for example, how does the gradient explode? (the dynamics of $\chi$). They can be derived via extensions of the techniques used by Yang and Schoenholz (2017) as well; intuitively any nonlinear function can be asymptotically approximated by linear ones, and each difference equation can be approximated by simple differential equations. The asymptotics of those differential equations can be shown to coincide with the asymptotics of the original difference equations.

Theoretically and empirically, WV does not change the mean dynamics of forward quantities like activation norm $\mathbf{p}$ but can be used to control gradient dynamics $\chi$. Intuitively, this is because each neuron at a width-changing layer "receives messages" from different numbers of neurons in the forward and backward computations. If for example $N^{(l)} = N^{(l-1)}/2$, then on the backward pass, the neuron receives half as many messages on the backward pass than in the forward, so we expect that its gradient should be half of what it would be when $N^{(l)} = N^{(l-1)}$. See Section 6.

On the other hand, VV will usually change both the forward and backward dynamics of mean field quantities. The phase transitions are many and complicated, but the overall trend is that, as we dampen the variance with depth, both forward and backward dynamics will dampen as well; the only major exception is weight gradients in ReLU resnets (see Appendix B.1). In contrast to WV which works the same for any nonlinearity, the phase diagram for VV is controlled by different quantities depending on the nonlinearity. We show through experiments that all of the complexities involved in VV theory are reflected in the practice of training neural networks: we can predict the contour lines of test time accuracy using only our mean field theory (Section 7 and Appendix B)

**Expressivity vs Trainability Tradeoff.** Yang and Schoenholz (2017) made the observation that the optimal initialization scheme for tanh resnets makes an optimal tradeoff between expressivity and trainability: if the initialization variances are too big, then the random network will suffer from gradient explosion with high probability; if they are too small, then the random network will be approximately constant (i.e. has low metric expressivity) with high probability. They posited that such tradeoff between expressivity and trainability in ReLU resnets is not observed because the gradient against weight parameters $w$ and $v$ are bounded w.r.t. depth (so that there is no gradient explosion)[7], while (metric) expressivity is exponential, thus strongly dominating the effect on final performance.

We confirm this behavior in tanh resnets when decaying their initialization variances with depth: When there is no decay, gradient explosion bottlenecks the test set accuracy after training; when we impose strong decay, gradient dynamics is mollified but now (metric) expressivity, being strongly constrained, bottlenecks performance (Section 7.2). Indeed, we can predict test set accuracy by level curves of gradient norm ratio $\chi_w^{(0)}/\chi_w^{(l)}$ in the region of small variance decay, while we can do the same with level curves of metric expressivity $\mathbf{s}$ when in the region of large decay (Fig. 3). The performance peaks at the intersection of these two regions.

---

[7]There is gradient explosions in the bias parameters $\chi_a$ and $\chi_b$ which follow the same dynamics as activation and metric expressivity dynamics $\mathbf{p}$ and $\mathbf{s}$. However, as performance increases when such explosions worsen, it seems that they do not bottleneck performance.

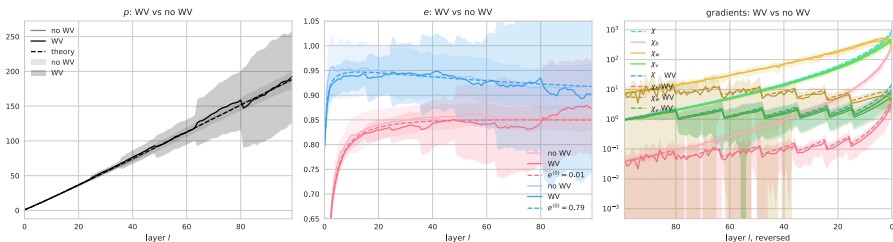

**Figure 1:** Using WV to control the gradient explosion of a tanh resnet. Shades indicate standard deviation (taken in normal scale, but possibly displayed in log scale), while solid lines indicate the corresponding mean. The left two plots show that mean forward dynamics are more or less preserved, albeit variance explodes toward the deeper layers, where WV is applied. The last plot show that the gradient dynamics is essentially suppressed to be a constant compared to the $\exp(\sqrt{L})$ dynamics of tanh resnet without width decay. Dashed lines indicate theoretical estimates in all three plot; solid, simulated data, which is generated from random residual networks with 100 layers and $N^{(0)} = 2048$, and we half the widths at layers $l = m^2$ for $m = 4, 5, \ldots, 9$.

Also corroborating Yang and Schoenholz (2017), we did not observe a tradeoff in ReLU resnets VV. In the regime with small to moderate variance decay, VV exerts its effect through metric expressivity, not gradient dynamics (Section 7.1) [8]. However, when we impose strong decay, gradient explosion, but not metric expressivity, predicts performance, in the unexpected way that worse gradient explosion correlates with better performance; that is, expressivity and trainability (as measured by gradient explosion) are both worsening, yet the performance increases! We currently have no explanation for this phenomenon but hope to find one in future work.

## 6  WIDTH VARIATION

Width variation is first passingly mentioned in Schoenholz et al. (2017) as a potential way to guide gradient dynamics for feedforward networks. We develop a complete mean field theory of WV for residual networks.

Via Table A.2 and Thm C.3, we see that width variation (WV) has two kinds of effects on the mean gradient norm: when compared to no width variation, it can multiply the squared gradient norm of biases $b_i$ or weights $w_{ij}$ by $N/M$ (which doesn't "stack", i.e. does not affect the squared gradient norm of lower layers), or it can multiply the squared gradient norm of $x_i$ by $N/\underline{N}$ (which "stacks", in the sense above, through the dynamics of $\chi$). We will focus on the latter "stacking" effect and assume $N^{(l)} = M^{(l)}$

Suppose from layer $l$ to layer $m$, $\boldsymbol{\chi}^{(m)}$ rises to $r$ times $\boldsymbol{\chi}^{(l)}$. If we vary the width so that $N^{(m-1)}$ is $rN^{(m)}$, then this gradient expansion is canceled, and $\boldsymbol{\chi}^{(m-1)} = (N^{(m)}/N^{(m-1)})(\sigma_v^2\sigma_w^2 l^{-\beta_v-\beta_w}\mathrm{V}\dot{\phi}(\mathbf{q}^{(m)}) + 1)\boldsymbol{\chi}^{(m)} = (\sigma_v^2\sigma_w^2 l^{-\beta_v-\beta_w}\mathrm{V}\dot{\phi}(\mathbf{q}^{(m)}) + 1)\boldsymbol{\chi}^{(l)}$ so that it is as if we restarted backpropagation at layer $m$.

Remarkably, changing the width does not change the mean field forward dynamics (for example the recurrences for $\mathbf{p}, \mathbf{q}, \boldsymbol{\gamma}, \boldsymbol{\lambda}$ remain the same) (Thm C.2). But, as always, if we reduce the width as part of WV (say, keeping $N^{(0)}$ the same but reducing the widths of later layers), the variance of the sampled dynamics will also increase; if we increase the width as part of WV (say, keeping $N^{(L)}$ the same but increasing the widths of earlier layers), the variance of the sampled dynamics will decrease.

We can apply this theory of WV to tanh residual networks ($\phi = \tanh$ in Table A.2) without VV. By Yang and Schoenholz (2017), tanh residual networks with all $\beta_\bullet = 0$ have gradient dynamics $\log(\boldsymbol{\chi}^{(m)}/\boldsymbol{\chi}^{(l)}) = \mathcal{A}(\sqrt{l} - \sqrt{m}) + \Theta(\log l - \log m)$ where $\mathcal{A} = \frac{4}{3}\sqrt{\frac{2}{\pi}}\frac{\sigma_v^2\sigma_w}{\sqrt{\sigma_v^2+\sigma_a^2}}$. If we place projection blocks projecting $N^{(l-1)} \mapsto N^{(l)}/\exp(\mathcal{A})$ at layer $l = n^2$, for $n = 1, 2, \ldots$, then the gradient norms would be bounded (above and below) across layers, as reasoned above. Indeed, this is what we see in Fig. 1. The rightmost subfigure compares, with log scale y-axis, the gradient

---

[8]As observed theoretically and empirically by Yang and Schoenholz (2017), there is gradient explosion in the bias parameters, but they do not seem to hold back performance



**Figure 2:** Zigzagging through $\beta_\bullet$-space with ReLU resnet. We trained a grid of ReLU residual networks, pinning all $\sigma_\bullet$s at 1, but varying the $\beta_\bullet$s as follows: The zig: fix $\beta_v = \beta_a = 0$; fix $\beta_w = \beta_b$ and increase both from 0 to 2 (making $\mathcal{V}_r = \beta_w + \beta_v$ go from 0 to 2 as well); The zag: fix $\beta_v = 0, \beta_w = \beta_b = 2$; increase $\beta_a$ from 0 to 2 (increasing $\mathcal{U}_r$ from 0 to 2 as well). For each setting of the hyperparameters, we train a network on MNIST with those hyperparameters for 30 epochs. We then report the accuracy that the network achieved on the training and test sets. The plots above are, in order from left to right, **(a) zig/test, (b) zag/test, (c) zig/train, (d) zag/train.** In the **zig**, we have overlaid a contour plot of **s** (computed from Thm C.2), which is almost identical to the contour plots of **p** and $\boldsymbol{\chi}^{(0)}/\boldsymbol{\chi}^{(l)}$; numbers indicate $\log 1 + \log \mathbf{s}$. The dashed line is given by $\frac{\mathcal{W}_r}{1-\mathcal{V}_r} l^{1-\mathcal{V}_r} = \mathcal{C}$ for $\mathcal{C} \approx 10$. In the **zag**, we have overlaid a contour plot of $\boldsymbol{\chi}_v^{(1)}/\boldsymbol{\chi}_v^{(l)}$ (computed from Thm C.3). Numbers indicate $\log \boldsymbol{\chi}_v^{(1)}/\boldsymbol{\chi}_v^{(l)}$.

dynamics with no WV to that with WV as described above. We see that our theory tracks the mean gradient dynamics remarkably precisely for both the WV and the no WV cases, and indeed, WV effectively caps the gradient norm for $l \geq 16$ (where WV is applied). The left two figures show the forward dynamics of **p** and **e**, and we see that the WV does not affect the mean dynamics as predicted by theory. However, we also see dramatic increase in deviation from the mean dynamics at every projection layer in the forward case. The backward dynamics (rightmost figure) similarly sees large deviations (1 standard deviation below mean is negative for $\chi_a$ and $\chi_w$), although the deviations for $\chi$ is more tamed but still much larger than without WV.

Therefore, width variation is unique in a few ways among all the techniques discussed in the mean field networks literature so far, including variance decay as studied below, adding skip connections, or changing activation functions: It can ameliorate or suppress altogether gradient explosion (or vanishing) problems without affecting the mean forward dynamics of **p**, **q**, $\boldsymbol{\lambda}$, $\boldsymbol{\gamma}$, **c**, **e**. To do so, it has to choose a trade-off from the following spectrum: At one end, we truncate neurons from the original network (say, keeping $N^{(0)}$ the same), so that we have fewer parameters, less compute, but larger deviations from the mean dynamics. At the other, we add neuron to the original network (say, keeping $N^{(L)}$ the same), so that we have more parameters, more compute, and smaller deviations from the mean dynamics.

## 7 VARIANCE VARIATION

### 7.1 RELU

A zigzag of parameters controls various asymptotics of ReLU resnet: "zigging" $\mathcal{V}_r := \min(\beta_v + \beta_b, \beta_a)$ from 0 to $> 1$, and then "zagging" $\mathcal{U}_r := \beta_v + \beta_w$ from 0 to $> 1$. During the zig, the asymptotics of **p** is subdued from $\exp(\text{poly}(l))$ to $\text{poly}(l)$. During the zag, it is further reduced to $\Theta(\log l)$ (at $\mathcal{U}_r = 1$) and $\Theta(1)$ (when $\mathcal{U}_r > 1$). On the contrary, the gradient dynamics of weight parameters become more explosive along the zigzag. During the zig, $\chi_w^{(1)}/\chi_w^{(l)}$ increases from $\Theta(1)$ to $\text{poly}(l)$ while $\chi_v^{(1)}/\chi_v^{(l)}$ increases from $\Theta(l^{\beta_v})$ to a bigger $\text{poly}(l)$. During the zig, both quantities increase the exponents of their polynomial dependence in $l$. In the interest of space, we stop here our sketch of VV dynamics for ReLU resnet, but refer the reader to Appendix B.1 for more detailed descriptions and Appendix C for proofs.

To test our theory, we sweep through these two macro-phases of the parameter space in our experiments and train an array of randomly initialized ReLU resnets; results are demonstrated in Fig. 2. The figure caption gives the experimental details involved. In addition, we provide heatmap and contour plots of various quantities of interest such as **p**, **e**, and $\chi_v^{(1)}/\chi_v^{(l)}$ in both the zig and the zag regimes in Fig. A.1 and Fig. A.2 respectively.

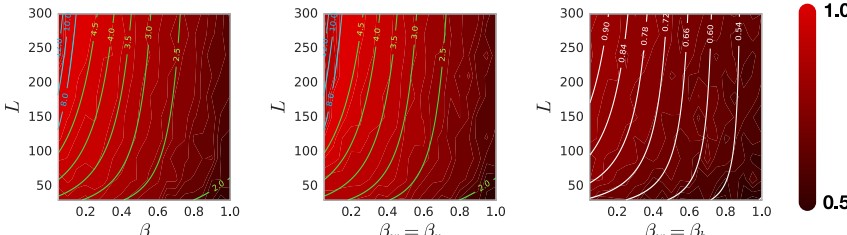

**Figure 3:** Sweeping through VV phases of tanh resnet. In all experiments here, we pin $\sigma_\bullet$s all to 1. From left to right: **(a)** and **(b)**. We sweep $\mathcal{U}_\mathrm{t} : 0 \nearrow 1$ in to ways, testing to what extent $\mathcal{U}_\mathrm{t}$ determines the final performance. In the first way (a), we set all $\beta_\bullet$s equal to a common $\beta$ and increase $\beta : 0 \nearrow 1$. In the second way (b), we clamp $\beta_b = \beta_a = 1$ and set $\beta_w = \beta_v$, increasing their common values from 0 to 1. The heatmaps are produced from final test set performance as in Fig. 2. As we can easily see, these two sweeps produce almost identical heatmaps. In both plots, there is a visible peak in the heatmaps in the upper left. On each of (a) and (b), we overlay the contours of $\chi_w^{(1)}/\chi_w^{(l)}$ (in blue) to the left of the peak and those of $\mathbf{p}$ (in green) to the right of the peak, the latter being very similar to those of $\mathbf{s}$. The blue numbers indicate $\log \chi_w^{(0)}/\chi_w^{(l)}$ while the green numbers indicate $\log \mathbf{p}$. **(c)**. We fix $\mathcal{U}_\mathrm{t} = 1$ by fixing $\beta_v = \beta_a = 1$ and sweep $\beta_w = \beta_b$ from 0 to 1, thereby sweeping $\mathcal{V}_\mathrm{t}$ from 1 to 2. The heatmap is obtained again with the same procedure as in Fig. 2 from the test set after training. Overlaid on top is a contour plot of $\mathbf{s}$, and the numbers indicate $\log \mathbf{s}$.

**Discussion.** First notice that training fails in the upper left corner of the zig. This happens because of numerical instability caused by exploding $\mathbf{p}$ and $\chi^{(0)}/\chi^{(l)}$, which grow like $\exp(\frac{\mathcal{W}_\mathrm{r}}{1-\mathcal{V}_\mathrm{r}} l^{1-\mathcal{V}_\mathrm{r}} + o(l^{1-\mathcal{V}_\mathrm{r}}))$ (Thms C.9 and C.11). Indeed, one of the contour lines of $\mathbf{p}$ traces out almost exactly where training fails. The dashed line is a level set of the dominant term of asymptotic expansion of $\log \mathbf{p}$, and we see it agrees with the contour of $\mathbf{p}$ very well. By increasing $\beta_w = \beta_b$, we effectively solved the activation explosion problem observed by Yang and Schoenholz (2017) without changing the activation function.

Second, observe that performance actually dips in the direction where $\chi^{(0)}/\chi^{(l)}$ decreases, quite counterintuitively[9]. This can be explained (as in Yang and Schoenholz (2017)) by noting that gradient against weights, $\chi_w$ and $\chi_v$, in fact respectively remain bounded and polynomial in $l$ (and changes rather mildly with $\mathcal{V}_\mathrm{r}$; see Fig. A.1); gradient against biases do experience the same behavior as $\chi$, but in general they are much less important than the weights, as parameters go. In addition, the performance is also dipping in the direction where $\mathbf{s}$ decreases (exponentially) in $(\mathcal{V}_\mathrm{r}, L)$-space. This is the quantity that essentially underlies the exponential expressivity result (as told from an extrinsic curvature perspective) of Poole et al. (2016); as $\mathbf{s}$ decreases dramatically, it gets harder and harder for a linear functional in the final layer to tell apart two input vectors. This exponential loss in expressivity dominates the effect on performance more than a polynomial reduction in gradient explosion.

Third, it is remarkable that in the zag regime, the level curves of $\chi_v^{(1)}/\chi_v^{(l)}$ (but not those of $\mathbf{p}, \mathbf{s}, \mathbf{e}, \chi$, or $\chi_w$!) accurately predict the contour of the test set performance, in such a counterintuitive way that greater gradient explosion $\chi_v^{(1)}/\chi_v^{(l)}$ correlates with better performance (Fig. 2(b)). Especially when $\beta_a$ (and thus also $\mathcal{U}_\mathrm{r}$) is large, the weight gradient dynamics are much more explosive than that of metric expressivity, so according to prevailing theory, gradient explosion should bottleneck performance, but instead the reality is the exact opposite. It is currently unclear if in certain situations like this, larger gradient expansion is actually beneficial, or if there is a quantity yet undiscovered which has the same level curves and can explain away this seeming paradox (like how $\mathbf{s}$ explains away $\chi^{(0)}/\chi^{(l)}$ above, in the zig regime). Of the quantities that appear in Fig. A.2, none foots this bill.

## 7.2 Tanh

As in the previous section, we briefly sketch the VV dynamics of tanh resnets, but defer a more detailed discussion to Appendix B.2 and the proofs to Appendix C. We are concerned with the scenario that $\mathbf{q} \to \infty$ with $l$, as otherwise higher layers become essentially unused by the network. The major phases of tanh resnet VV dynamics are then determined by when $\mathcal{U}_t := \min(\beta_v, \beta_a) < 1$ and when $\mathcal{U}_t = 1$ (Thms C.5 and C.6). Within the former, the gradient dynamics is controlled by $\mathcal{W}_t := 1 - \beta_w + \mathcal{U}_t - 2\beta_v$; as $\mathcal{W}_t$ starts positive, decrease to 0, and then becomes negative, the gradient ratio $\chi^{(0)}/\chi^{(l)}$ starts out as $\exp(\mathrm{poly}(l))$, turns into polynomial, and finally becomes bounded. When $\mathcal{U}_t = 1$, $\chi^{(0)}/\chi^{(l)}$ is always subpolynomial, with $\mathcal{V}_t := \beta_v + \beta_w$ making it bounded as $\mathcal{V}_t$ increases past 1. On the other hand, the dynamics of $\mathbf{p}$ is quite simple, with $\mathbf{p} = \Theta(l^{1-\mathcal{U}_t})$ when $\mathcal{U}_t < 1$ and $\mathbf{p} = \Theta(\log l)$ when $\mathcal{U}_t = 1$.

This theory enables us to predict and optimize an VV initialization scheme for tanh resnets. We sweep through the the two phases described above, train the corresponding random networks, and exhibit the results in Fig. 3. The figure caption details the experimental setup.

**Discussions.** Fig. 3(a) and Fig. 3(b) sweep through $\mathcal{U}_t$ from 0 to 1 in two different ways while obtaining almost identical test set performance heatmap, showing indeed that the hyperparameters $\beta_\bullet$ exert their effect through $\mathcal{U}_t = \min(\beta_v, \beta_a)$. In each of these two plots, peak accuracies happen in the upper left. To the left of such a peak, gradient norm $\chi_w^{(1)}/\chi_w^{(l)}$ predicts the accuracy, while to the right of such a peak, metric expressivity $\mathbf{s}$ (and in this case $\mathbf{p}$ as well because they induce similar contours) does. But $\chi_w^{(1)}/\chi_w^{(l)}$ would not do well in the large $\beta$ region because the slopes of its contours are too steep; conversely $\mathbf{s}$ would not predict well in the small $\beta$ region because its contour slopes are not steep enough. Indeed, one sees that the slopes of the heatmap level set boundaries decrease as the accuracy levels increase (and as $\beta$ decreases from 1), but when the level peaks, the slope suddenly becomes much steeper (compare the left boundary of the peak to its right boundary). Our observation here reaffirms the trainability vs expressivity tradeoff studied in Yang and Schoenholz (2017).

In Fig. 3(c), we study the $\mathcal{U}_t = 1$ phase. Here $\mathbf{s}$ alone predicts performance (though in the large $\beta_w = \beta_b$ region, final accuracy becomes more random and the prediction is not as great). This is expected, as $\chi_\bullet^{(0)}/\chi_\bullet^{(l)}$ is now subpolynomial for all $\bullet = v, w, a, b$ (Thm C.6), so that trainability is not an issue.

## 8 Conclusion

In this paper, we derived the mean field theory of width and variance variation and showed that they are powerful methods to control forward (VV) and backward (VV + WV) dynamics. We proved that even with a fixed architecture and activation function, the mean field dynamics of a residual neural network can still be manipulated at will by these two methods. Extraordinarily, the mean field theory we developed allowed us to accurately predict the performances of trained MNIST models relative to different initializations, but one puzzling aspect remains where test set accuracy seems to increase as gradient explosion worsens in one regime of random ReLU resnets.

**Open Problems.** We solved a small part, width variation, of the program to construct mean field theories of state-of-the-art neural networks used in practice. Many open problems still remain, and the most important of them include but is not limited to 1. batchnorm, 2. convolution layers, and 3. recurrent layers. In addition, more work is needed to mathematically justify our "physical" assumptions Axiom 1 and Axiom 2 to a "math" problem. We hope readers will take note and contribute toward deep mean field theory.

---

[9]Even though Fig. 2(a) has overlaid on top the contours of $\mathbf{s}$, they are similar enough to those of $\chi^{(0)}/\chi^{(l)}$ that we are identifying them here.

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

**Table A.1:** Glossary of Symbols. "Mean normalized" is abbreviated "m.n."

| Symbol | Meaning | Ref |
|---|---|---|
| $\sigma_\bullet$ | standard deviation of trainable parameter $\bullet$ | |
| $x^{(l)}$ | activation vector/input vector | |
| $h^{(l)}$ | hidden vector | |
| $N^{(l)}$ | width of layer $l$ activation $x^{(l)}$ | |
| $M^{(l)}$ | width of layer $l$ hidden vector $h^{(l)}$ | |
| $\mathbf{p}^{(l)}$ | m.n. squared length of activation vector $x^{(l)}$ | 4.1 |
| $\mathbf{q}^{(l)}$ | m.n. squared length of hidden vector $h^{(l)}$ | 4.1 |
| $\boldsymbol{\gamma}^{(l)}$ | m.n. dot product $x^{(l)} \cdot x^{(l)\prime}$ | 4.2 |
| $\boldsymbol{\lambda}^{(l)}$ | m.n. dot product $h^{(l)} \cdot h^{(l)\prime}$ | 4.2 |
| $\mathbf{s}^{(l)}$ | m.n. squared distance $\|x^{(l)} - x^{(l)\prime}\|^2$ | 4.2 |
| $\mathbf{e}^{(l)}$ | cosine distance $\boldsymbol{\gamma}^{(l)}/\sqrt{\mathbf{p}^{(l)}\mathbf{p}^{(l)\prime}}$ | 4.2 |
| $\mathbf{e}^*$ | limit value of $\mathbf{e}^{(l)}$ as $l \to \infty$ | |
| $\mathbf{c}^{(l)}$ | cosine distance $\boldsymbol{\lambda}^{(l)}/\sqrt{\mathbf{q}^{(l)}\mathbf{q}^{(l)\prime}}$ | 4.2 |
| $\boldsymbol{\chi}^{(l)}$ | m.n. gradient squared norm w.r.t. $x^{(l)}$ | 4.3 |
| $\boldsymbol{\chi}_\bullet^{(l)}$ | m.n. gradient squared norm w.r.t. trainable parameter $\bullet$ | 4.3 |
| $\phi$ | variable nonlinearity $\mathbb{R} \to \mathbb{R}$ | |
| V | variance integral transform | C.1 |
| W | covariance integral transform | C.1 |
| $\mathcal{U}_\mathrm{t}$ | $\min(\beta_v, \beta_a)$; controls tanh resnet VV dynamics | C.4 |
| $\mathcal{W}_\mathrm{t}$ | $1 - \beta_w + \mathcal{U}_\mathrm{t} - 2\beta_v$; controls tanh resnet VV dynamics | C.4 |
| $\mathcal{V}_\mathrm{t}$ | $\beta_v + \beta_w$; controls tanh resnet VV dynamics | C.4 |
| $\mathcal{V}_\mathrm{r}$ | $\beta_v + \beta_w$ controls ReLU resnet VV dynamics | C.8 |
| $\mathcal{U}_\mathrm{r}$ | $\min(\beta_v + \beta_b, \beta_a)$; controls ReLU resnet VV dynamics | C.8 |
| $\mathcal{W}_\mathrm{r}$ | $\frac{1}{2}\sigma_v^2\sigma_w^2$; controls ReLU resnet VV dynamics | C.8 |

**Table A.2:** Mean field dynamics with VV and WV

| Forward (Thm C.2) | Backward (Thm C.3) |
|---|---|
| $\mathbf{q} = \sigma_w^2 l^{-\beta_w}\underline{\mathbf{p}} + \sigma_b^2 l^{-\beta_b}$ $\Delta\mathbf{p} = \sigma_v^2 l^{-\beta_v}\mathrm{V}\phi(\mathbf{q}) + \sigma_a^2 l^{-\beta_a}$ $\boldsymbol{\lambda} = \sigma_w^2 l^{-\beta_w}\underline{\boldsymbol{\lambda}} + \sigma_b^2 l^{-\beta_b}$ $\Delta\boldsymbol{\gamma} = \sigma_v^2 l^{-\beta_v}\mathrm{W}\phi(\mathbf{q},\boldsymbol{\lambda}) + \sigma_a^2 l^{-\beta_a}.$ | $\underline{\boldsymbol{\chi}}/\boldsymbol{\chi} = \begin{cases} (N/\underline{N})(\sigma_v^2\sigma_w^2 l^{-\beta_v-\beta_w}\mathrm{V}\dot\phi(\mathbf{q}) + 1) \\ (\sigma_v^2\sigma_w^2 l^{-\beta_v-\beta_w}\mathrm{V}\dot\phi(\mathbf{q}) + 1) \end{cases}$ $\boldsymbol{\chi}_b = (N/M)\sigma_v^2 l^{-\beta_v}\boldsymbol{\chi}\mathrm{V}\dot\phi(\mathbf{q})$ $\boldsymbol{\chi}_w = (N/M)\sigma_v^2 l^{-\beta_v}\boldsymbol{\chi}\mathrm{V}\dot\phi(\mathbf{q})\underline{\mathbf{p}}$ $\boldsymbol{\chi}_v = \boldsymbol{\chi}\mathrm{V}\phi(\mathbf{q}) \quad \boldsymbol{\chi}_a = \boldsymbol{\chi} \quad \boldsymbol{\chi}_\pi = \boldsymbol{\chi}\underline{\mathbf{p}}.$ |

The two cases for $\underline{\boldsymbol{\chi}}/\boldsymbol{\chi}$ are resp. for a projection and a normal residual block, assuming $\sigma_\pi = 1$. The V and W operators are defined in Defn C.1.

# Appendices

## A  MEAN FIELD ASSUMPTIONS

We make several key "mean field" assumptions, which were formulated in their entirety first in Yang and Schoenholz (2017) (though Axiom 2(a) has been stated first by Schoenholz et al. (2017)). While these assumptions may be mathematically unsatisfying, identifying them and discovering that they lead to highly precise prediction is in fact one of the most important contributions of deep mean field theory.

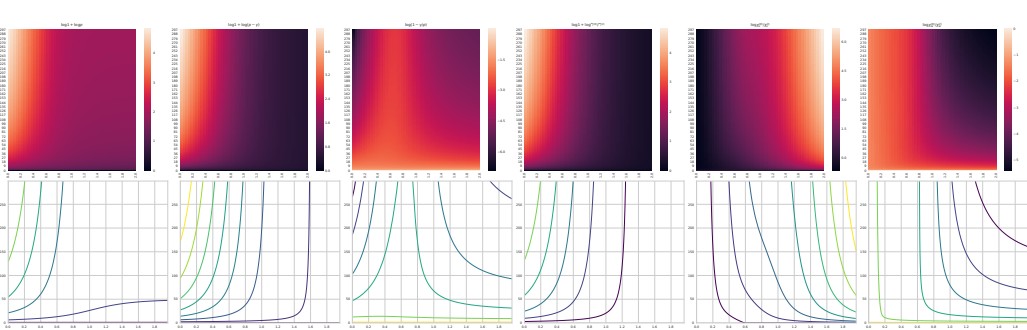

**Figure A.1:** Heatmap and contour plots of various quantities in the "zig" regime of Section 7.1. Note that we squashed things like **p** with large range so that the colors are meaningful.

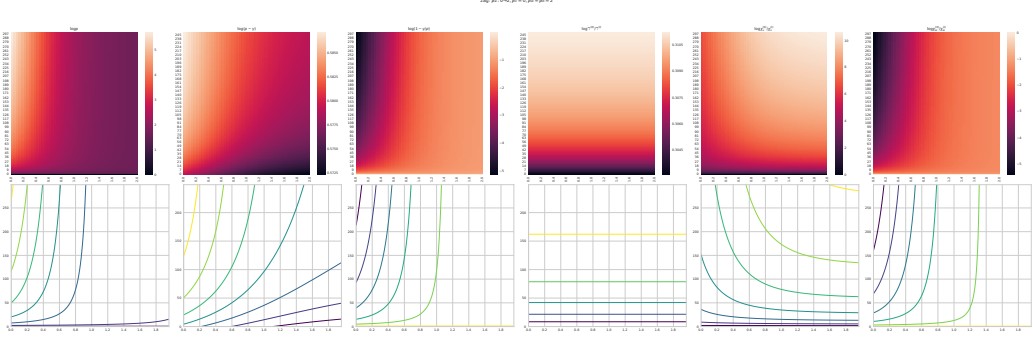

**Figure A.2:** Heatmap and contour plots of various quantities in the "zag" regime of Section 7.1. Note that we squashed things like **p** with large range so that the colors are meaningful.

**Axiom 1** (Symmetry of activations and gradients). *(a) We assume* $\langle (h_i^{(l)})^2 \rangle = \langle (h_j^{(l)})^2 \rangle$ *and* $\langle (x_i^{(0)})^2 \rangle = \langle (x_j^{(0)})^2 \rangle$ *for any* $i, j, l$. *(b) We also assume that the gradient* $\partial E / \partial x_i^{(l)}$ *with respect to the loss function* $E$ *satisfies* $\langle (\partial E / \partial x_i^{(l)})^2 \rangle = \langle (\partial E / \partial x_j^{(l)})^2 \rangle$ *for any* $i, j, l$.

**Axiom 2** (Gradient independence). *(a) We assume the we use a different set of weights for backpropagation than those used to compute the network outputs, but sampled i.i.d. from the same distributions. (b) For any loss function* $E$, *we assume that the gradient at layer* $l$, $\partial E / \partial x_i^{(l)}$, *is independent from all activations* $h_j^{(l)}$ *and* $x_j^{(l-1)}$ *from the previous layer.*

One can see that Axiom 1(a) is satisfied if the input $x^{(0)} \in \{\pm 1\}^N$ and Axiom 1(b) is satisfied if Axiom 2 below is true and the gradient at the last layer $\partial E / \partial xL \in \{\pm 1\}^N$. Axiom 2(a) was first made in Schoenholz et al. (2017) for computing the mean field theory of gradients for feedforward tanh networks. This is similar to the practice of feedback alignment (Lillicrap et al., 2016).

## B    OVERVIEW: DYNAMICS UNDER VARIANCE VARIATION

As discuss in Section 6, WV is essentially a post-hoc technique to tweak an existing gradient dynamic without changing the forward dynamics. Thus in this section we assume all widths are constant, $N^{(l)} = N^{(m)} = M^{(n)}$ for any $m, l, n$, so that WV can be applied as a "touch-up" if necessary. We will overview the phases and transitions due to VV, but defer all proofs to later sections.

### B.1    RESIDUAL NETWORK WITH RELU

It was shown in Yang and Schoenholz (2017) that, with no variance decay, in an ReLU resnet, both the mean squared activation norm ($\mathbf{p}$) and the mean squared gradient norm ($\boldsymbol{\chi}$) explode exponentially with depth, and this causes training to fail for even 100 layer networks. We show that this problem is in fact extremely easy to fix, requiring no architectural changes at all, only that $\beta_\bullet$ be increased from 0 so that the randomization variances decay across depth (Thms C.9 and C.11).

**Gradient quantities.**    The main driver of this gradient mollification is $\mathcal{V}_r := \beta_v + \beta_w$. When $\mathcal{V}_r \in [0, 1)$, the gradient norm varies like $\boldsymbol{\chi}^{(0)} / \boldsymbol{\chi}^{(l)} = \exp(\Theta(l^{1-\mathcal{V}_r}))$; when $\mathcal{V}_r = 0$ this recapitulate the exponential behavior derived in Yang and Schoenholz (2017). When $\mathcal{V}_r = 1$, it experiences a sharp phase transition, where now $\boldsymbol{\chi}^{(0)} / \boldsymbol{\chi}^{(l)} = \mathrm{poly}(l)$. As $\mathcal{V}_r$ becomes larger than 1, $\boldsymbol{\chi}^{(0)} / \boldsymbol{\chi}^{(l)} = \Theta(1)$, all bounded! Fig. B.3 verifies this result empirically, and in fact show that our computed asymptotic expansions in Thm C.11 are highly accurate predictors. It is both interesting and important to note that the gradient norms for actual trainable parameters, such as $\boldsymbol{\chi}_w$ and $\boldsymbol{\chi}_v$, are affected differently by $\mathcal{V}_r$. In fact, $\boldsymbol{\chi}_w^{(1)} / \boldsymbol{\chi}_w^{(l)}$ is *bounded*(!) with $l$ when $\mathcal{V}_r < 1$ (the $\mathcal{V}_r = 0$ case is already observed in Yang and Schoenholz (2017)) but phase transitions to $\mathrm{poly}(l)$ for $\mathcal{V}_r \geq 1$, while $\boldsymbol{\chi}_v^{(1)} / \boldsymbol{\chi}_v^{(l)}$ is already $\mathrm{poly}(l)$ when $\mathcal{V}_r < 1$, and remains so as $\mathcal{V}_r$ is increased to $> 1$. Curiously, greater gradient explosion in $\boldsymbol{\chi}_v$ predicts better performance in the $\mathcal{V}_r > 1$ regime, and we currently do not know if this is intrinsic or there are confounding variables; see Section 7.1.

**Length quantities.**    Similarly, $\mathcal{V}_r$ is the primary conduit for mollifying the behavior of squared activation norms $\mathbf{p}$ and $\mathbf{q}$ (Thm C.9). Like the gradient dynamics, $\mathbf{p} = \exp(\Theta(l^{1-\mathcal{V}_r}))$ when $\mathcal{V}_r < 1$; when $\mathcal{V}_r = 0$ this recapitulates the results of Yang and Schoenholz (2017). As $\mathcal{V}_r$ rise to 1 and above, $\mathbf{p}$ experiences a phase transition into polynomial dynamics, but unlike the case of $\boldsymbol{\chi}$, it is not constant when $\mathcal{V}_r > 1$. Instead, a different parameter, $\mathcal{U}_r := \min(\beta_v + \beta_b, \beta_a)$ drives the asymptotics of $\mathbf{p}$ in the $\mathcal{V}_r > 1$ regime. When $\mathcal{U}_r \in [0, 1)$ is small, $\mathbf{p}$ grows like $\Theta(l^{1-\mathcal{U}_r})$. The instant $\mathcal{U}_r$ hits 1, $\mathbf{p}$ is just logarithmic, $\mathbf{p} = \Theta(\log l)$. As $\mathcal{U}_r$ shoots past 1, $\mathbf{p}$ becomes constant. Thus the dynamics of $\mathbf{p}$ is governed by a zigzag through $(\beta_\bullet)_{\bullet=v,w,a}$ space. Fig. B.4 goes through each of the five cases discussed above and verifies that our asymptotics are correct.

**Cosine distance.**    $\mathbf{e} = \boldsymbol{\gamma} / \mathbf{p}$ measures how well the input space geometry (angles, in this case) is preserved as the input space is propagated through each layer. Its dynamics are much simpler than those of $\mathbf{p}$ and $\boldsymbol{\chi}$ above. If $\mathbf{e}^{(0)} = 1$, then $\mathbf{e}^{(l)} = 1$ for all $l$ trivially. If $\mathbf{e}^{(0)} < 1$, then we have one of the following two cases, • If $\mathcal{V}_r \leq 1$ or $\mathcal{U}_r \leq 1$, then $\mathbf{e} \to 1$ irrespective of initial data $\mathbf{p}^{(0)}$ and $\boldsymbol{\gamma}^{(0)}$.

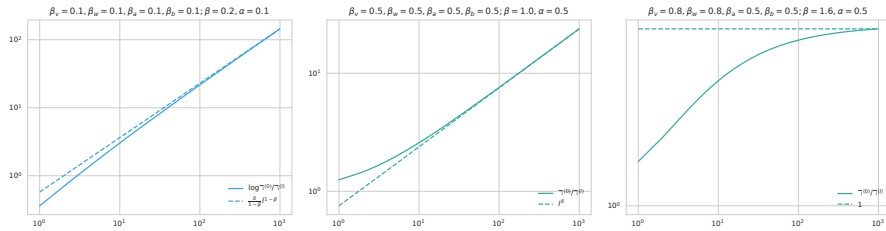

**Figure B.3:** We verify the asymptotic characterizations of $\chi$ given by Thm C.11. Here solid lines indicate computations done via the recurrences Table A.2, while dashed lines indicate asymptotics proved in Thm C.11. To facilitate visualization, all green (but not blue) dashed lines are shifted vertically to match the end point of the corresponding solid lines. From left to right: **(a)** log-log plot of the *log* of $\chi^{(0)}/\chi^{(l)}$, against the predicted leading term $\frac{\mathcal{W}_\mathrm{r}}{1-\mathcal{V}_\mathrm{r}}l^{1-\mathcal{V}_\mathrm{r}}$, when $\mathcal{V}_\mathrm{r} = .2$; thus $\chi^{(0)}/\chi^{(l)}$ is superpolynomial. As in Thm C.11, $\mathcal{W}_\mathrm{r} = \frac{1}{2}\sigma_v^2\sigma_w^2$. **(b)** log-log plot of $\chi^{(0)}/\chi^{(l)}$ when $\mathcal{V}_\mathrm{r} = 1$, showing it is polynomial. **(c)** $\chi^{(0)}/\chi^{(l)}$ is bounded above when $\mathcal{V}_\mathrm{r} = 1.6$.

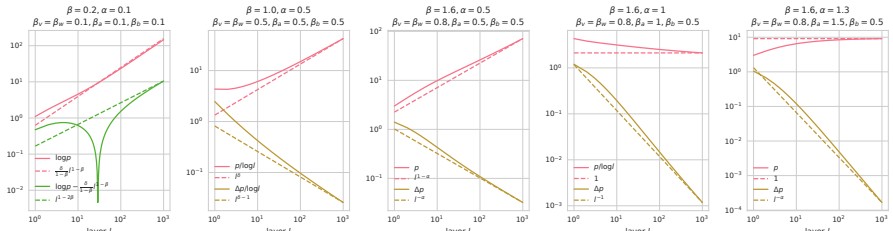

**Figure B.4:** We verify the asymptotic characterizations of **p** given by Thm C.9. Here solid lines indicate computations done via the recurrences Table A.2, while dashed lines indicate asymptotics proved in Thm C.9. In all but the leftmost plot, we show both **p** and $\Delta\mathbf{p} = \mathbf{p} - \underline{\mathbf{p}}$ (possibly adjusted for log factors) and their asymptotics. To facilitate visualization, all dashed lines except the red one in the leftmost plot are shifted vertically to match the end point of the corresponding solid lines. In the leftmost plot, the red lines are respectively $\log\mathbf{p}$ (solid) and the leading term $\frac{\mathcal{W}_\mathrm{r}}{1-\mathcal{V}_\mathrm{r}}l^{1-\mathcal{V}_\mathrm{r}}$ in its asymptotic expansion (dashed). The green lines in the same plot are the remainder (solid) and its polynomial asymptotic form, ignoring the leading coefficient (dashed). From left to right gives one example for each of the 5 cases discussed in the main text: **(a)** $\mathcal{V}_\mathrm{r} < 1$ and $\mathbf{p} = \exp(\mathrm{poly}(l))$. **(b)** $\mathcal{V}_\mathrm{r} = 1$ and **p** phase transitions into polynomial dynamics. **(c)** $\mathcal{V}_\mathrm{r} > 1, \mathcal{U}_\mathrm{r} < 1$ and $\mathbf{p} = \mathrm{poly}(l)$. **(d)** $\mathcal{V}_\mathrm{r} > 1, \mathcal{U}_\mathrm{r} = 1$ and $\mathbf{p} = \Theta(\log l)$. **(e)** $\mathcal{V}_\mathrm{r} > 1, \mathcal{U}_\mathrm{r} > 1$ and **p** becomes bounded but $\Delta\mathbf{p} = \Theta(l^{-\mathcal{U}_\mathrm{r}})$.

• If $\mathcal{V}_\mathrm{r} > 1$ and $\mathcal{U}_\mathrm{r} > 1$, then **e** converges to a fixed point $\mathbf{e}^* < 1$ which depends on the initial data $\mathbf{p}^{(0)}$ and $\boldsymbol{\gamma}^{(0)}$, at a rate of $\Theta(l^{1-\mathcal{U}_\mathrm{r}})$. Thus ReLU *very much* likes to collapse the input space into a single point ($\mathbf{e} = 1$ means every two input vectors get mapped to the same output vector), and the only way to prevent this from happening is to make the $\beta_\bullet$s so high, that higher layers barely modifies the computation done by lower layers at all. Indeed, the second condition $\mathcal{V}_\mathrm{r} > 1$ and $\mathcal{U}_\mathrm{r} > 1$ ensures that $\mathbf{p} = \Theta(1)$ as discussed above (Thm C.9), so that as $l \to \infty$, layer $l$'s residual adds to $x^{(l-1)}$ only a vector of vanishing size compared to the size of $x^{(l-1)}$.

## B.2    RESIDUAL NETWORK WITH TANH

While ReLU resnet depends heavily on $\mathcal{V}_\mathrm{r} = \beta_v + \beta_w$ and $\mathcal{U}_\mathrm{r} = \min(\beta_v + \beta_b, \beta_a), \mathcal{U}_\mathrm{t} := \min(\beta_v, \beta_a)$ and $\mathcal{W}_\mathrm{t} := 1 - \beta_w + \mathcal{U}_\mathrm{t} - 2\beta_v$ are the key quantities determining the dynamics in the case of tanh resnet, with $\mathcal{V}_\mathrm{t} = \beta_v + \beta_w = \mathcal{V}_\mathrm{r}$ playing a minor role. We will study tanh resnets in the setting where $\mathbf{q} \to \infty$ as $l \to \infty$; otherwise, **p** is bounded, meaning that higher layers become essentially unused by the neural network (similar to the discussion made in Appendix B.1(Cosine distance) above). In this setting, it can be shown that $\mathcal{U}_\mathrm{t} \leq 1$ (Thm C.5).

**Gradient quantities.** By Thm C.6, as long as $\mathcal{U}_\mathrm{t}$ stays below 1, the asymptotics of $\chi$ is entirely governed by $\mathcal{W}_\mathrm{t}$, which is 1 when all $\beta_\bullet$s are 0 and most of the time decreases as $\beta_\bullet$s are increased.

When $\mathcal{W}_t > 0$, $\chi^{(0)}/\chi^{(l)} = \exp(\Theta(l^{\frac{1}{2}\mathcal{W}_t}))$; the results of Yang and Schoenholz (2017) are recovered by setting all $\beta_\bullet$s to 0, thus $\mathcal{W}_t = 1$ and $\chi^{(0)}/\chi^{(l)} = \exp(\Theta(l^{\frac{1}{2}}))$. When $\mathcal{W}_t$ hits 0, $\chi^{(0)}/\chi^{(l)}$ becomes polynomial, and as $\mathcal{W}_t$ dips below 0, gradient expansion becomes bounded.

If $\mathcal{U}_t = 1$, $\chi^{(0)}/\chi^{(l)}$ is automatically suppressed to be subpolynomial. The only minor phase transition here is going from $\mathcal{V}_r = 1$ to $\mathcal{V}_r > 1$ (and $\mathcal{V}_r$ cannot be less than 1 by our assumption that $\mathbf{q} \to \infty$). In the former case, the gradient expansion is $\exp(\Theta(\sqrt{\log l}))$, while in the latter case it is bounded.

**Length quantities** have simpler asymptotics determined by $\mathcal{U}_t$: either $\mathcal{U}_t < 1$ and $\mathbf{p} = \Theta(l^{1-\mathcal{U}_t})$, or $\mathcal{U}_t = 1$ and $\mathbf{p} = \Theta(\log l)$ (Thm C.5). **Cosine distance,** unlike the case of ReLU resnets, can be controlled effectively by $\beta_a$ and $\beta_v$ (Thm C.7). When $\beta_a > \beta_v$, the magnitude of $a_i^{(l)}$ drops much more quickly with depth than that of $v_{ij}^{(l)}$, so that higher layers experience the chaotic phase (Schoenholz et al., 2017; Yang and Schoenholz, 2017), driving $\mathbf{e}^{(l)}$ toward the limit point $\mathbf{e}^* = 0$. On the other end, when $\beta_a < \beta_v$, $v_{ij}^{(l)}$ vanishes in comparison to $a_i^{(l)}$ with large $l$, so that the higher layers experience the stability phase (Schoenholz et al., 2017; Yang and Schoenholz, 2017), collapsing all inputs to the same output vector, sending $\mathbf{e}^{(l)} \to 1$. Only when $\beta_a = \beta_v$ could the fixed point $\mathbf{e}^*$ be controlled explicitly by $\sigma_v$ and $\sigma_a$, with $\mathbf{e}^*$ given by the equation $\mathbf{e}^* = \frac{\sigma_v^2 \frac{2}{\pi} \arcsin(\mathbf{e}^*) + \sigma_a^2}{\sigma_v^2 + \sigma_a^2}$ as in the case with no VV (Yang and Schoenholz, 2017).

## C  MAIN THEOREMS

**Asymptotic notations.**  The expressions $f = O(g) \iff g = \Omega(f)$ have their typical meanings, and $f = \Theta(g)$ iff $f = O(g), g = O(f)$. We take $f(x) = \tilde{O}(g(x)) \iff g(x) = \tilde{\Omega}(f(x))$ to mean $f(x) = O(g \log^k x)$ for some $k \in \mathbb{Z}$ (this is slightly different from the standard usage of $\tilde{O}$), and $f = \tilde{\Theta}(g) \iff f = \tilde{O}(g) \ \& \ g = \tilde{O}(f)$. All asymptotic notations are sign-less, i.e. can indicate either positive or negative quantities, unless stated otherwise.

We recall integral transforms from Yang and Schoenholz (2017):

**Definition C.1.** Define the transforms V and W by $\mathrm{V}\phi(\rho) := \mathrm{E}[\phi(z)^2 : z \sim \mathcal{N}(0, \rho)]$ and $\mathrm{W}\phi(\rho, \nu) := \mathrm{E}[\phi(z)\phi(z') : (z, z') \sim \mathcal{N}(0, \begin{pmatrix} \rho & \nu \\ \nu & \rho \end{pmatrix})]$.

Table A.2 summarizes the derived recurrences.

### C.1  RECURRENCES

Yang and Schoenholz (2017) gave recurrences for mean field quantities $\mathbf{p}, \mathbf{q}, \boldsymbol{\gamma}, \boldsymbol{\lambda}, \boldsymbol{\chi}$ under the assumption of constant initialization variances across depth. The proofs there carry over straightforwardly when variance varies from layer to layer. Schoenholz et al. (2017) also derived backward dynamics for when the width of the network is not constant. Generalizing to the residual network case requires some careful justifications of independences, so we provide proof for gradient dynamics; but we omit the proof for the forward dynamics as it is not affected by nonconstant width and is almost identical to the constant variance case.

**Theorem C.2.** *For any nonlinearity $\phi$ in an FRN, regardless of whether widths vary across layers,*

$$\mathbf{q}^{(l)} = \sigma_w^2 l^{-\beta_w} \mathbf{p}^{(l-1)} + \sigma_b^2 l^{-\beta_b}$$
$$\mathbf{p}^{(l)} = \sigma_v^2 l^{-\beta_v} \mathrm{V}\phi(\mathbf{q}^{(l)}) + \sigma_a^2 l^{-\beta_a} + \mathbf{p}^{(l-1)}$$
$$\boldsymbol{\lambda}^{(l)} = \sigma_w^2 l^{-\beta_w} \boldsymbol{\lambda}^{(l-1)} + \sigma_b^2 l^{-\beta_b}$$
$$\boldsymbol{\gamma}^{(l)} = \sigma_v^2 l^{-\beta_v} \mathrm{W}\phi(\mathbf{q}^{(l)}, \boldsymbol{\lambda}^{(l)}) + \sigma_a^2 l^{-\beta_a} + \boldsymbol{\gamma}^{(l-1)}.$$

**Theorem C.3.** *Suppose a random residual network receives a fixed gradient vector $\left(\partial E/\partial x_i^{(L)}\right)_{i=1}^{N^{(l)}}$ with respect to some cost function $E$, at its last layer. For any nonlinearity $\phi$ in an FRN, under*

*Axiom 1 and Axiom 2, whenever $\dot{\phi}(\zeta)^2$ has finite variance for any Gaussian variable $\zeta$,*

$$\underline{\chi} = \begin{cases} (N/\underline{N})(\sigma_v^2\sigma_w^2 l^{-\beta_v-\beta_w}\mathrm{V}\dot{\phi}(\mathbf{q}) + \sigma_\pi^2)\chi & \text{if layer } l \text{ has projection connection} \\ (\sigma_v^2\sigma_w^2 l^{-\beta_v-\beta_w}\mathrm{V}\dot{\phi}(\mathbf{q}) + 1)\chi & \text{if layer } l \text{ has identity connection} \end{cases}$$

$$\chi_b = (N/M)\sigma_v^2 l^{-\beta_v}\chi\mathrm{V}\dot{\phi}(\mathbf{q})$$

$$\chi_w = (N/M)\sigma_v^2 l^{-\beta_v}\chi\mathrm{V}\dot{\phi}(\mathbf{q})\underline{\mathbf{p}}$$

$$\chi_v = \chi\mathrm{V}\phi(\mathbf{q})$$

$$\chi_a = \chi$$

$$\chi_\pi = \chi\underline{\mathbf{p}}.$$

*Proof.* We will show the proof for the projection connection case; the identity connection case is similar but easier.

Write $\eta_i^{(l)} = \frac{\partial E}{\partial x_i^{(l)}}$. We have the following derivative computations:

$$\frac{\partial x_i}{\partial \underline{x}_j} = \pi_{ij} + \sum_k v_{ik}\dot{\phi}(h_k)\frac{\partial h_k}{\partial \underline{x}_j}, \quad \frac{\partial x_i}{\partial h_j} = v_{ij}\dot{\phi}(h_j), \quad \frac{\partial h_i}{\partial \underline{x}_j} = w_{ij}, \quad \frac{\partial h_i}{\partial w_{ij}} = \underline{x}_j, \quad \frac{\partial h_i}{\partial b_i} = 1,$$

$$\frac{\partial x_i}{\partial v_{ik}} = \phi(h_k), \quad \frac{\partial x_i}{\partial a_i} = 1, \quad \frac{\partial x_i}{\partial \pi_{ij}} = \underline{x}_j$$

Then

$$\underline{\eta}_j = \sum_{i=1}^N \eta_i(\pi_{ij} + \sum_{k=1}^M v_{ik}\dot{\phi}(h_k)\frac{\partial h_k}{\partial \underline{x}_j})$$

$$= \sum_{i=1}^N \eta_i(\pi_{ij} + \sum_{k=1}^M v_{ik}\dot{\phi}(h_k)w_{kj})$$

Thus,

$$\langle\underline{\eta}_j^2\rangle = \langle[\sum_{i=1}^N \eta_i(\pi_{ij} + \sum_{k=1}^M v_{ik}\dot{\phi}(h_k)w_{kj})]^2\rangle$$

$$= \langle\sum_{i=1}^N \eta_i^2[(\pi_{ij} + \sum_{k=1}^M v_{ik}\dot{\phi}(h_k)w_{kj})]^2\rangle$$

$$+ 2\sum_{i<i'}\langle\eta_i\eta_{i'}(\pi_{ij} + \sum_{k=1}^M v_{ik}\dot{\phi}(h_k)w_{kj})(\pi_{i'j} + \sum_{k=1}^M v_{i'k}\dot{\phi}(h_k)w_{kj})\rangle$$

$$= N\langle\eta_1^2\rangle(\langle\pi_{11}^2\rangle + \langle[\sum_{k=1}^M v_{1k}\dot{\phi}(h_k)w_{kj}]^2\rangle)$$

$$+ 2\sum_{i<i'}\langle\eta_i\eta_{i'}\rangle\langle(\pi_{ij} + \sum_{k=1}^M v_{ik}\dot{\phi}(h_k)w_{kj})(\pi_{i'j} + \sum_{k=1}^M v_{i'k}\dot{\phi}(h_k)w_{kj})\rangle$$

where in the second equality, we expanded algebraically, and in the third equality, we use the symmetry assumption Axiom 1 and the independence assumption Axiom 2. Now, $\langle[\sum_{k=1}^M v_{1k}\dot{\phi}(h_k)w_{kj}]^2\rangle = \sum_{k=1}^M\langle v_{1k}^2\dot{\phi}(h_k)^2 w_{kj}^2\rangle = \sum_{k=1}^M \frac{\sigma_v^2 l^{-\beta_v}}{M}\mathrm{V}\dot{\phi}(\mathbf{q})\frac{\sigma_w^2 l^{-\beta_w}}{N} = \sigma_v^2 l^{-\beta_v}\mathrm{V}\dot{\phi}(\mathbf{q})\frac{\sigma_w^2 l^{-\beta_w}}{N}$ by the independence of $\{v_{ik}\}_{i,k} \cup \{h_k\}_k \cup \{w_{kj}\}_{k,j}$ (by our independence assumptions Axiom 2). Similarly, because $\{\pi_{ij}\}_j \cup \{\pi_{i'j}\}_j \cup \{v_{ik}\}_k \cup \{v_{i'k}\}_k$ for $i \neq i'$ is mutually independent by our assumptions, one can easily see that $\langle(\pi_{ij} + \sum_{k=1}^M v_{ik}\dot{\phi}(h_k)w_{kj})(\pi_{i'j} + \sum_{k=1}^M v_{i'k}\dot{\phi}(h_k)w_{kj})\rangle = 0$.

Therefore,

$$\underline{\chi} = N\chi(\frac{\sigma_\pi^2}{\underline{N}} + \sigma_v^2 l^{-\beta_v} V\dot\phi(\mathbf{q})\frac{\sigma_w^2 l^{-\beta_w}}{\underline{N}})$$
$$= \frac{N}{\underline{N}}\chi(\sigma_\pi^2 + \sigma_v^2\sigma_w^2 l^{-\beta_v-\beta_w} V\dot\phi(\mathbf{q})).$$

For the other gradients, we have (where we apply Axiom 2 implicitly)

$$\frac{\partial E}{\partial b_j} = \sum_{i=1}^N \frac{\partial E}{\partial x_i}\frac{\partial x_i}{\partial h_j}\frac{\partial h_j}{\partial b_j} = \sum_{i=1}^N \eta_i v_{ij}\dot\phi(h_j)$$

$$\chi_b = \langle[\sum_{i=1}^N \eta_i v_{ij}\dot\phi(h_j)]^2\rangle = N\chi\frac{\sigma_v^2 l^{-\beta_v}}{M}V\dot\phi(\mathbf{q}) = \frac{N}{M}\sigma_v^2 l^{-\beta_v}\chi V\dot\phi(\mathbf{q});$$

$$\frac{\partial E}{\partial w_{ji}} = \sum_{i=1}^N \frac{\partial E}{\partial x_i}\frac{\partial x_i}{\partial h_j}\frac{\partial h_j}{\partial w_{ji}} = \sum_{i=1}^N \eta_i v_{ij}\dot\phi(h_j)\underline{x}_j$$

$$\chi_w = \langle\sum_{i=1}^N \eta_i v_{ij}\dot\phi(h_j)\underline{x}_j\rangle = N\chi\frac{\sigma_v^2 l^{-\beta_v}}{M}V\dot\phi(\mathbf{q})\underline{\mathbf{p}} = \frac{N}{M}\sigma_v^2 l^{-\beta_v}\chi V\dot\phi(\mathbf{q})\underline{\mathbf{p}}$$

$$\frac{\partial E}{\partial v_{ik}} = \frac{\partial E}{\partial x_i}\frac{\partial x_i}{\partial v_{ik}} = \eta_i\phi(h_k) \implies \chi_v = \langle[\eta_i\phi(h_k)]^2\rangle = \chi V\phi(\mathbf{q})$$

$$\frac{\partial E}{\partial a_i} = \frac{\partial E}{\partial x_i}\frac{\partial x_i}{\partial a_i} = \frac{\partial E}{\partial x_i} \implies \chi_a = \chi$$

$$\frac{\partial E}{\partial \pi_{ij}} = \frac{\partial E}{\partial x_i}\frac{\partial x_i}{\partial \pi_{ij}} = \eta_i\underline{x}_j \implies \chi_\pi = \chi\underline{\mathbf{p}}$$

$\square$

## C.2 RESIDUAL NETWORKS WITH TANH

In this section we derive the asymptotics of various mean field quantities for tanh resnet. The main proof technique is to bound the dynamics in question with known dynamics of difference equations (as in Yang and Schoenholz (2017)).

**Definition C.4.** Let $\mathcal{U}_t := \min(\beta_v, \beta_a)$, $\mathcal{W}_t := 1 - \beta_w + \mathcal{U}_t - 2\beta_v$, $\mathcal{V}_t := \beta_v + \beta_w$.

**Theorem C.5.** *Suppose $\phi = \tanh$, and $\mathbf{q}^{(l)} \to \infty$ as $l \to \infty$.*

1. *If $\mathcal{U}_t < 1$, then $1 > \beta_w + \mathcal{U}_t$ and*

$$\mathbf{p}^{(l)} = \Theta(l^{1-\mathcal{U}_t})$$
$$\mathbf{q}^{(l)} = \Theta(l^{1-\beta_w-\mathcal{U}_t}).$$

*More specifically,*

$$(\mathbf{p}^{(l)}, \mathbf{q}^{(l)}) \sim \begin{cases} (\sigma_v^2 l^{1-\beta_v}, \sigma_v^2\sigma_w^2 l^{1-\beta_v-\beta_w}) & \text{if } \beta_v < \beta_a \\ (\sigma_a^2 l^{1-\beta_a}, \sigma_a^2\sigma_w^2 l^{1-\beta_a-\beta_w}) & \text{if } \beta_a < \beta_v \\ ((\sigma_a^2 + \sigma_v^2)l^{1-\beta_v}, (\sigma_a^2 + \sigma_v^2)\sigma_w^2 l^{1-\beta_v-\beta_w}) & \text{if } \beta_a = \beta_v. \end{cases}$$

2. *If $\mathcal{U}_t = 1$, then $\beta_w = 0$, and $\mathbf{p}^{(l)} = \Theta(\log l) = \mathbf{q}^{(l)}$. More specifically,*

$$(\mathbf{p}^{(l)}, \mathbf{q}^{(l)}) \sim \begin{cases} (\sigma_v^2, \sigma_w^2\sigma_v^2 \log l) & \text{if } \beta_v < \beta_a \\ (\sigma_a^2 \log l, \sigma_w^2\sigma_a^2 \log l) & \text{if } \beta_v > \beta_a \\ ((\sigma_a^2 + \sigma_v^2)\log l, \sigma_w^2(\sigma_a^2 + \sigma_v^2)\log l) & \text{if } \beta_v = \beta_a. \end{cases}$$

3. $\mathcal{U}_t$ *cannot be greater than 1.*

*Proof.* **Claim 1.** We have $V\phi(\mathbf{q}) = 1 - \sqrt{\frac{2}{\pi}}\mathbf{q}^{-1/2} + \Theta(\mathbf{q}^{-3/2})$ by Lemma D.5. Thus

$$\mathbf{p} - \underline{\mathbf{p}} = \sigma_v^2 l^{-\beta_v}(1 - \sqrt{2/\pi}\mathbf{q}^{-1/2} + \Theta(\mathbf{q}^{-3/2})) + \sigma_a^2 l^{-\beta_a}$$
$$= \sigma_v^2 l^{-\beta_v} + \sigma_a^2 l^{-\beta_a} - o(l^{-\beta_v})$$
$$\mathbf{p} \sim \begin{cases} \sigma_v^2 l^{1-\beta_v} & \text{if } \beta_v < \beta_a \\ \sigma_a^2 l^{1-\beta_a} & \text{if } \beta_a < \beta_v \\ (\sigma_a^2 + \sigma_v^2) l^{1-\beta_v} & \text{if } \beta_a = \beta_v \end{cases}$$

where we used the assumption $1 - \mathcal{U}_t > 0$. Thus $\mathbf{p}^{(l)} = \Theta(l^{1-\mathcal{U}_t})$ and goes to infinity with $l$.

Then $\mathbf{q} = \sigma_w^2 l^{-\beta_w}\mathbf{p} + \sigma_b^2 l^{-\beta_b}$. Because we assumed $\mathbf{q} \to \infty$, the first term necessarily dominates the second, and $1 - \mathcal{U}_t - \beta_w > 0$. The possible asymptotics of $\mathbf{q}$ are then

$$\mathbf{q} \sim \begin{cases} \sigma_v^2 \sigma_w^2 l^{1-\beta_v-\beta_w} & \text{if } \beta_v < \beta_a \\ \sigma_a^2 \sigma_w^2 l^{1-\beta_a-\beta_w} & \text{if } \beta_v > \beta_a \\ (\sigma_a^2 + \sigma_v^2)\sigma_w^2 l^{1-\beta_v-\beta_w} & \text{if } \beta_a = \beta_v \end{cases}$$

which gives $\mathbf{q}^{(l)} = \Theta(l^{1-\beta_w-\mathcal{U}_t})$.

**Claim 2.** If $\mathcal{U}_t = 1$, then

$$\mathbf{p} \sim \begin{cases} \sigma_v^2 \log l & \text{if } \beta_v < \beta_a \\ \sigma_a^2 \log l & \text{if } \beta_v > \beta_a \\ (\sigma_a^2 + \sigma_v^2) \log l & \text{if } \beta_v = \beta_a. \end{cases}$$

Then for $\mathbf{q}$ to go to infinity, $\beta_w$ has to be 0, so that $\mathbf{q} = \Theta(\log l)$ as well, and

$$\mathbf{q} \sim \begin{cases} \sigma_w^2 \sigma_v^2 \log l & \text{if } \beta_v < \beta_a \\ \sigma_w^2 \sigma_a^2 \log l & \text{if } \beta_v > \beta_a \\ \sigma_w^2 (\sigma_a^2 + \sigma_v^2) \log l & \text{if } \beta_v = \beta_a. \end{cases}$$

**Claim 3.** If $\mathcal{U}_t > 1$, then $\mathbf{p} = \Theta(1) \implies \mathbf{q} = \Theta(1)$, a contradiction. $\qquad\square$

**Theorem C.6.** *Suppose $\phi = \tanh$, and $\mathbf{q}^{(l)} \to \infty$ with $l$. Recall $\mathcal{W}_t = 1 - \beta_w + \mathcal{U}_t - 2\beta_v$ and $\mathcal{V}_t = \beta_v + \beta_w$. Then $\log(\chi^{(0)}/\chi^{(l)})$ is*

*If $\mathcal{U}_t < 1$*

    *If $\mathcal{W}_t > 0$*
        $\Theta(l^{\frac{1}{2}\mathcal{W}_t})$
    *If $\mathcal{W}_t = 0$*
        $\Theta(\log l)$
    *If $\mathcal{W}_t < 0$*
        $\Theta(1)$

*If $\mathcal{U}_t = 1$*

    *If $\mathcal{V}_t < 1$*
        *This case cannot happen*
    *If $\mathcal{V}_t = 1$*
        $\Theta(\sqrt{\log l})$
    *If $\mathcal{V}_t > 1$*
        $\Theta(1)$

*Proof.* By Thm C.3 and Lemma D.4, we have

$$\log(\underline{\chi}/\chi) = \log(1 + \sigma_v^2 \sigma_w^2 l^{-\beta_v - \beta_w} \mathrm{V}\dot{\phi}(\mathbf{q}))$$

$$= \log(1 + \sigma_v^2 \sigma_w^2 l^{-\beta_v - \beta_w}(C\mathbf{q}^{-1/2} + \Theta(\mathbf{q}^{-3/2})))$$

$$= \sigma_v^2 \sigma_w^2 l^{-\beta_v - \beta_w} C\mathbf{q}^{-1/2} + \Theta(l^{-2\beta_v - 2\beta_w})$$

where $C = \frac{2}{3}\sqrt{\frac{2}{\pi}}$. Since $\mathbf{q}^{(l)}$ has different growth rates depending on the hyperparameters, we need to consider different cases:

- If $\mathcal{U}_t < 1$, then Thm C.5 implies $\beta_w + \mathcal{U}_t < 1$ and $\mathbf{q} = \Theta(l^{1-\beta_w-\mathcal{U}_t})$. So $l^{-\beta_v-\beta_w}\mathbf{q}^{-1/2} = l^{-\beta_v-\beta_w}\Theta(l^{-\frac{1}{2}(1-\beta_w-\mathcal{U}_t)}) = \Theta(l^{\frac{1}{2}(-1-\beta_w+\mathcal{U}_t-2\beta_v)})$.

  - If $1 - \beta_w + \mathcal{U}_t - 2\beta_v > 0$, $\log \chi^{(0)}/\chi^{(l)} = \sum_{m=1}^{l} \Theta(m^{\frac{1}{2}(-1-\beta_w+\mathcal{U}_t-2\beta_v)}) = \Theta(l^{\frac{1}{2}(1-\beta_w+\mathcal{U}_t-2\beta_v)})$.
  - If $1 - \beta_w + \mathcal{U}_t - 2\beta_v = 0$, then $l^{-\beta_v-\beta_w}\mathbf{q}^{-1/2} = \Theta(l^{-1})$. Thus $\log \chi^{(0)}/\chi^{(l)} = \sum_{m=1}^{l} \Theta(m^{-1}) = \Theta(\log l)$.
  - If $1 - \beta_w + \mathcal{U}_t - 2\beta_v < 0$, then $\frac{1}{2}(-1 - \beta_w + \mathcal{U}_t - 2\beta_v) > 1$, so that $\log \chi^{(0)}/\chi^{(l)} = \Theta(1)$.

- If $\mathcal{U}_t = 1$, then by Thm C.5, $\beta_w = 0$ and $\mathbf{q} = \Theta(\log l)$. So $l^{-\beta_v-\beta_w}\mathbf{q}^{-1/2} = \Theta(l^{-\beta_v-\beta_w}/\sqrt{\log l})$.

  - If $\beta_v + \beta_w < 1$, then $\beta_v < 1 \implies \mathcal{U}_t < 1$, contradiction.
  - If $\beta_v + \beta_w = 1$, then $\log \chi^{(0)}/\chi^{(l)} = \sum_{m=1}^{l} \Theta(l^{-1}/\sqrt{\log l}) = \Theta(\sqrt{\log l})$.
  - If $\beta_v + \beta_w > 1$, then $\log \chi^{(0)}/\chi^{(l)} = \Theta(1)$.

$\square$

**Theorem C.7.** *Suppose $\phi = \tanh$ and $\mathbf{q} \to \infty$. If $\mathbf{e}^{(0)} < 1$, then $\mathbf{e}^{(l)}$ converges to a fixed point $\mathbf{e}^*$, given by the equations*

$$\mathbf{e}^* = \begin{cases} 0 & \text{if } \beta_a > \beta_v \\ \frac{\sigma_v^2 \frac{2}{\pi} \arcsin(\mathbf{e}^*) + \sigma_a^2}{\sigma_v^2 + \sigma_a^2} & \text{if } \beta_a = \beta_v \\ 1 & \text{if } \beta_a < \beta_v. \end{cases}$$

Note that in the case $\beta_a = \beta_v$, we recover the fixed point of tanh residual network without variance decay.

*Proof.* We have

$$\sigma_v^2 l^{-\beta_v} \mathrm{W}\phi(\mathbf{q}, \mathbf{cq}) + \sigma_a^2 l^{-\beta_a} = \boldsymbol{\gamma} - \underline{\boldsymbol{\gamma}}$$

$$= \mathbf{ep} - \underline{\mathbf{ep}}$$

$$= \mathbf{ep} - \underline{\mathbf{e}}\mathbf{p} + \underline{\mathbf{e}}\mathbf{p} - \underline{\mathbf{ep}}$$

$$= (\mathbf{e} - \underline{\mathbf{e}})\mathbf{p} + \underline{\mathbf{e}}(\mathbf{p} - \underline{\mathbf{p}})$$

$$= (\underline{\mathbf{e}} + (\mathbf{e} - \underline{\mathbf{e}})\frac{\mathbf{p}}{\mathbf{p} - \underline{\mathbf{p}}})(\mathbf{p} - \underline{\mathbf{p}})$$

$$\frac{\sigma_v^2 l^{-\beta_v} \mathrm{W}\phi(\mathbf{q}, \mathbf{cq}) + \sigma_a^2 l^{-\beta_a}}{\sigma_v^2 l^{-\beta_v} \mathrm{V}\phi(\mathbf{q}) + \sigma_a^2 l^{-\beta_a}} - \underline{\mathbf{e}} = (\mathbf{e} - \underline{\mathbf{e}})\frac{\mathbf{p}}{\mathbf{p} - \underline{\mathbf{p}}}$$

Using Lemma D.1 and Lemma D.5, we can see that the LHS is monotonic (increasing or decreasing) for large enough $l$. Therefore $\mathbf{e}^{(l)}$ is a bounded monotonic sequence for large enough $l$, a fortiori it has a fixed point.

If we express $\mathbf{e} = \mathbf{e}^* + \epsilon$, $\boldsymbol{\gamma}^{(l)} = (\mathbf{e}^* + \epsilon^{(l)})\mathbf{p}^{(l)}$, then

$$\frac{\sigma_v^2 l^{-\beta_v} \mathrm{W}\phi(\mathbf{q}, \mathbf{cq}) + \sigma_a^2 l^{-\beta_a}}{\sigma_v^2 l^{-\beta_v} \mathrm{V}\phi(\mathbf{q}) + \sigma_a^2 l^{-\beta_a}} = \mathbf{e}^* + \underline{\epsilon} + (\epsilon - \underline{\epsilon})\frac{\mathbf{p}}{\mathbf{p} - \underline{\mathbf{p}}}$$

**RHS** It's easy to verify via Thm C.5 that either $\underline{\mathbf{p}}/(\mathbf{p}-\underline{\mathbf{p}}) = \Theta(l \log l)$ (when $\mathcal{U}_{\mathrm{t}} = 1$) or $\mathbf{p}/(\mathbf{p}-\underline{\mathbf{p}}) = \Theta(l)$ (all other cases). If $\epsilon - \underline{\epsilon} = \Omega((l \log l)^{-1})$, then $\epsilon = \Omega(\log \log l)$ (by Euler-MacLaurin formula), which would be a contradiction to $\epsilon = o(1)$. Thus $(\epsilon - \underline{\epsilon})\frac{\mathbf{p}}{\mathbf{p}-\underline{\mathbf{p}}} = o(1)$. [10] Hence the RHS goes to 0 with $l$.

**LHS** If $\beta_v > \beta_a$, then the LHS converges to 1. So $\mathbf{e}^* = 1$.

If $\beta_v < \beta_a$, then the LHS $\sim \frac{\mathrm{W}\phi(\mathbf{q},\mathbf{cq})}{\mathrm{V}\phi(\mathbf{q})}$. As $l \to \infty$, $\mathbf{c} \sim \mathbf{e} \to \mathbf{e}^*$, and $\mathrm{W}\phi(\mathbf{q}, \mathbf{e}^*\mathbf{q}) \to \frac{2}{\pi}\arcsin(\mathbf{e}^*)$, and $\mathrm{V}\phi(\mathbf{q}) \to 1$. Therefore, $\mathbf{e}^* = \frac{2}{\pi}\arcsin(\mathbf{e}^*)$, for which there are 2 solutions, 0, the stable fixed point, and 1, the unstable fixed point. In particular, for all $\mathbf{e}^{(0)} < 1$, $\mathbf{e}^{(l)} \to 0$.

If $\beta_v = \beta_a$, then taking limits $l \to \infty$, we get

$$\frac{\sigma_v^2 \frac{2}{\pi}\arcsin(\mathbf{e}^*) + \sigma_a^2}{\sigma_v^2 + \sigma_a^2} = \mathbf{e}^*.$$

$\square$

### C.3 RESIDUAL NETWORK WITH RELU

The asymptotics of ReLU resnet depends on the following values:

**Definition C.8.**

$$\mathcal{V}_{\mathrm{r}} := \beta_v + \beta_w, \qquad \mathcal{U}_{\mathrm{r}} := \min(\beta_v + \beta_b, \beta_a) = \beta_v + \min(\beta_b, \beta_a - \beta_v), \qquad \mathcal{W}_{\mathrm{r}} := \frac{1}{2}\sigma_v^2 \sigma_w^2$$

**Theorem C.9.** *Let $\phi = \mathrm{ReLU}$. Then we have the following asymptotics of $\mathbf{p}$ and $\mathbf{q}$:*

- *If $\mathcal{V}_{\mathrm{r}} > 1$*
  - *If $\mathcal{U}_{\mathrm{r}} \in [0, 1)$*
    - $\mathbf{p} = \Theta(l^{1-\mathcal{U}_{\mathrm{r}}})$ *and* $\mathbf{q} = \Theta(l^{\max(1-\mathcal{U}_{\mathrm{r}}-\beta_w, -\beta_b)})$.
  - *If $\mathcal{U}_{\mathrm{r}} = 1$*
    - $\mathbf{p} = \Theta(\log l)$ *and* $\mathbf{q} = \begin{cases} \Theta(l^{-\beta_w} \log l) & \text{if } \beta_w \le \beta_b \\ \Theta(l^{-\beta_b}) & \text{otherwise.} \end{cases}$
  - *If $\mathcal{U}_{\mathrm{r}} > 1$*
    - $\mathbf{p} = \Theta(1)$, $\mathbf{p} - \underline{\mathbf{p}} = \Theta(l^{-\mathcal{U}_{\mathrm{r}}})$ *and* $\mathbf{q} = \Theta(l^{-\min(\beta_w, \beta_b)})$.

- *If $\mathcal{V}_{\mathrm{r}} = 1$*
  - *If $\mathcal{W}_{\mathrm{r}} \neq 1 - \mathcal{U}_{\mathrm{r}}$*
    - $\mathbf{p} = \Theta(l^{\max(\mathcal{W}_{\mathrm{r}}, 1-\mathcal{U}_{\mathrm{r}})})$, *and* $\mathbf{q} = \Theta(l^{\max(\max(\mathcal{W}_{\mathrm{r}}, 1-\mathcal{U}_{\mathrm{r}})-\beta_w, -\beta_b)})$.
  - *Otherwise*
    - $\mathbf{p} = \Theta(l^{\mathcal{W}_{\mathrm{r}}} \log l)$ *and* $\mathbf{q} = \begin{cases} \Theta(l^{\mathcal{W}_{\mathrm{r}}-\beta_w} \log l) & \text{if } \beta_b \ge \beta_w - \mathcal{W}_{\mathrm{r}} \\ \Theta(l^{-\beta_b}) & \text{otherwise} \end{cases}$

- *If $\mathcal{V}_{\mathrm{r}} < 1$*

---

[10] In fact, since $\epsilon$ must be $o(\log \log \cdots \log l)$ for any chain of logs, $\epsilon - \underline{\epsilon} = o((l \log l \log \log l \cdots \log^{(k)} l)^{-1})$ for any $k$, where $\log^{(k)}$ is $k$-wise composition of $\log$; so $(\epsilon - \underline{\epsilon})\frac{\mathbf{p}}{\mathbf{p}-\underline{\mathbf{p}}} = o((\log \log l \cdots \log^{(k)} l)^{-1})$ for any $k$.

- ○ $\mathbf{p}, \mathbf{q} = \exp(\frac{\mathcal{W}_\mathrm{r}}{1-\mathcal{V}_\mathrm{r}} l^{1-\mathcal{V}_\mathrm{r}} + \tilde{\Theta}(l^{\max(0,1-2\mathcal{V}_\mathrm{r})}))$. *In particular,* $\mathbf{p} = \exp(\frac{\mathcal{W}_\mathrm{r}}{1-\mathcal{V}_\mathrm{r}} l^{1-\mathcal{V}_\mathrm{r}} + R(l; \mathcal{W}_\mathrm{r}, \mathcal{V}_\mathrm{r}) + O(1))$, $R$ *is the same* $R$ *as in Lemma D.7, depending on only* $l$*,* $\mathcal{W}_\mathrm{r}$*, and* $\mathcal{V}_\mathrm{r}$*, with*

$$R = \begin{cases} \Theta(l^{1-2\mathcal{V}_\mathrm{r}}) & \text{if } \mathcal{V}_\mathrm{r} < 1/2 \\ \Theta(\log l) & \text{if } \mathcal{V}_\mathrm{r} = 1/2 \\ \Theta(1) & \text{if } \mathcal{V}_\mathrm{r} > 1/2. \end{cases}$$

*and* $\mathbf{q} = \exp(\frac{\mathcal{W}_\mathrm{r}}{1-\mathcal{V}_\mathrm{r}} l^{1-\mathcal{V}_\mathrm{r}} + R(l; \mathcal{W}_\mathrm{r}, \mathcal{V}_\mathrm{r}) - \beta_w \log l + O(1))$.

*In particular,* $\mathbf{p}, \mathbf{q} = \mathrm{poly}(l)$ *if* $\mathcal{V}_\mathrm{r} \geq 1$.

*Proof.* Since $\mathrm{V}\phi(\mathbf{q}) = \frac{1}{2}\mathbf{q}$, we have

$$\begin{aligned}
\mathbf{p} - \underline{\mathbf{p}} &= \sigma_v^2 l^{-\beta_v} \mathrm{V}\phi(\mathbf{q}) + \sigma_a^2 l^{-\beta_a} \\
&= \frac{1}{2}\sigma_v^2 l^{-\beta_v}(\sigma_w^2 l^{-\beta_w}\underline{\mathbf{p}} + \sigma_b^2 l^{-\beta_b}) + \sigma_a^2 l^{-\beta_a} \\
&= \frac{1}{2}\sigma_v^2\sigma_w^2 l^{-\beta_v-\beta_w}\underline{\mathbf{p}} + \frac{1}{2}\sigma_v^2\sigma_b^2 l^{-\beta_v-\beta_b} + \sigma_a^2 l^{-\beta_a}.
\end{aligned}$$

We apply Lemma D.8 with $\beta = \beta_v + \beta_w$, $\delta = \frac{1}{2}\sigma_v^2\sigma_w^2$, and $\alpha = \min(\beta_v + \beta_b, \beta_a) = \beta_v + \min(\beta_b, \beta_a - \beta_v)$ to upper and lower bound our dynamics here, with appropriately chosen $C$. With a little bit of case work, we obtain the desired results. $\square$

**Theorem C.10.** *Let* $\phi = \mathrm{ReLU}$. *Suppose* $\mathbf{e}^{(0)} < 1$.

- *If* $\mathcal{V}_\mathrm{r} \leq 1$ *or* $\mathcal{U}_\mathrm{r} \leq 1$, *then* $\lim_{l\to\infty} \mathbf{e}^{(l)} = 1$.

- *If* $\mathcal{V}_\mathrm{r} > 1$ *and* $\mathcal{U}_\mathrm{r} > 1$, *then* $\mathbf{e}^{(l)}$ *converges to a fixed point* $\mathbf{e}^* < 1$, *dependent on the initial data* $\mathbf{p}^{(0)}$ *and* $\boldsymbol{\gamma}^{(0)}$, *at a rate of* $\Theta(l^{-\mathcal{U}_\mathrm{r}+1})$.

*Proof.* By Cho and Saul (2009); Yang and Schoenholz (2017), we have $\mathrm{W}\phi(\mathbf{q}, \mathbf{cq}) = \mathrm{V}\phi(\mathbf{q})\mathbb{J}_1(\mathbf{c}) = \frac{1}{2}\mathbf{q} \cdot \frac{1}{\pi}(\sqrt{1-\mathbf{c}^2} + (\pi - \arccos(\mathbf{c}))\mathbf{c})$. As in the proof of Thm C.7, we have

$$\frac{\sigma_v^2 l^{-\beta_v}\mathbf{q}/2\mathbb{J}_1(\mathbf{c}) + \sigma_a^2 l^{-\beta_a}}{\sigma_v^2 l^{-\beta_v}\mathbf{q}/2 + \sigma_a^2 l^{-\beta_a}} - \underline{\mathbf{e}} = (\mathbf{e} - \underline{\mathbf{e}})\frac{\mathbf{p}}{\mathbf{p} - \underline{\mathbf{p}}}$$

But $\frac{\sigma_v^2 l^{-\beta_v}\mathbf{q}/2\mathbb{J}_1(\mathbf{c})+\sigma_a^2 l^{-\beta_a}}{\sigma_v^2 l^{-\beta_v}\mathbf{q}/2+\sigma_a^2 l^{-\beta_a}} \geq \mathbb{J}_1(\mathbf{c}) \geq \mathbf{c} \geq \underline{\mathbf{e}}$, where the last inequality holds for all large $l$, by Lemma D.1. So the LHS is nonnegative for large $l$, which implies $\mathbf{e}^{(l)}$ is a bounded monotonically increasing sequence for large enough $l$s and thus has a limit $\mathbf{e}^*$.

Writing $\mathbf{e} = \mathbf{e}^* + \epsilon$, we have

$$\mathbf{e}^* + \underline{\epsilon} + (\epsilon - \underline{\epsilon})\frac{\mathbf{p}}{\mathbf{p} - \underline{\mathbf{p}}} = \frac{\sigma_v^2 l^{-\beta_v}\mathbf{q}/2\mathbb{J}_1(\mathbf{c}) + \sigma_a^2 l^{-\beta_a}}{\sigma_v^2 l^{-\beta_v}\mathbf{q}/2 + \sigma_a^2 l^{-\beta_a}} \tag{$\star$}$$

**Suppose** $\mathcal{V}_\mathrm{r} \leq 1$ **or** $\mathcal{U}_\mathrm{r} \leq 1$.

**LHS.** By Thm C.9, $\mathbf{p} = \omega(1)$ (assuming $\sigma_v, \sigma_w > 0$), so that $\mathbf{p}/(\mathbf{p} - \underline{\mathbf{p}}) = \tilde{O}(l)$. As in the proof of Thm C.7, $\epsilon - \underline{\epsilon}$ cannot be $\tilde{\Omega}(l^{-1})$, or else $\epsilon \to \infty$. Thus $(\epsilon - \underline{\epsilon})\mathbf{p}/(\mathbf{p} - \underline{\mathbf{p}}) = o(1)$, and the LHS in the limit $l \to \infty$ becomes $\mathbf{e}^*$.

**RHS.** In all cases, we will find $\mathbf{e}^* = 1$.

If $\mathbf{q} = o(l^{\beta_v - \beta_a})$, in the RHS $\sigma_a^2 l^{-\beta_a}$ dominates, and the RHS converges to 1.

If $\mathbf{q} = \omega(l^{\beta_v - \beta_a})$, then in the RHS $\frac{1}{2}\sigma_v^2 l^{-\beta_v}\mathbf{q}$ dominates. By Lemma D.1:

- If $\mathbf{p} = \omega(l^{\beta_w - \beta_b})$, then $\mathbf{c} \sim \mathbf{e} \to \mathbf{e}^*$, so that in the limit $l \to \infty$, $\mathbf{e}^* = \mathbb{J}_1(\mathbf{e}^*)$. This equation has solution $\mathbf{e}^* = 1$.

- If $\mathbf{p} = o(l^{\beta_w - \beta_b})$, then $\mathbf{c} \to 1$, so that $\mathbf{e}^* = \mathbb{J}_1(1) = 1$.

- If $\mathbf{p} \sim Cl^{\beta_w - \beta_b}$, then $\mathbf{c} \to \frac{\sigma_w^2 C \mathbf{e}^* + \sigma_b^2}{\sigma_w^2 C + \sigma_b^2}$, and we have an equation $\mathbf{e}^* = \mathbb{J}_1\left(\frac{\sigma_w^2 C \mathbf{e}^* + \sigma_b^2}{\sigma_w^2 C + \sigma_b^2}\right)$. Note that since $\mathbf{e}^* \in [0, 1]$, $\frac{\sigma_w^2 C \mathbf{e}^* + \sigma_b^2}{\sigma_w^2 C + \sigma_b^2} = \frac{\sigma_w^2 C}{\sigma_w^2 C + \sigma_b^2} \cdot \mathbf{e}^* + \frac{\sigma_b^2}{\sigma_w^2 C + \sigma_b^2} \cdot 1 \ge \mathbf{e}^*$, with equality iff $\mathbf{e}^* = 1$. Since $\mathbb{J}_1$ is monotonic, $\mathbf{e}^* = \mathbb{J}_1\left(\frac{\sigma_w^2 C \mathbf{e}^* + \sigma_b^2}{\sigma_w^2 C + \sigma_b^2}\right)$ iff the equality condition above is satisifed, i.e. $\mathbf{e}^* = 1$.

If $\mathbf{q} \sim Dl^{\beta_w - \beta_b}$ for some $D$, then the RHS is

$$\frac{\frac{1}{2}\sigma_v^2 D \mathbb{J}_1(\mathbf{c}) + \sigma_a^2}{\frac{1}{2}\sigma_v^2 D + \sigma_a^2} + o(1)$$

by the same logic as in the proof of Lemma D.1. As in the above case, Lemma D.1 yields the following results:

- If $\mathbf{p} = \omega(l^{\beta_w - \beta_b})$, then $\mathbf{c} \sim \mathbf{e} \to \mathbf{e}^*$, so that in the limit $l \to \infty$, $\mathbf{e}^* = \frac{\frac{1}{2}\sigma_v^2 D \mathbb{J}_1(\mathbf{e}^*) + \sigma_a^2}{\frac{1}{2}\sigma_v^2 D + \sigma_a^2}$. Since the RHS of this equation is a convex combination of $\mathbb{J}_1(\mathbf{e}^*)$ and 1, and $\mathbb{J}_1(\mathbf{e}^*) \ge \mathbf{e}^*$ by the monotonicity of $\mathbb{J}_1$, the equality can hold only if $\mathbb{J}_1(\mathbf{e}^*) = \mathbf{e}^*$. The only such $\mathbf{e}^*$ is 1.

- If $\mathbf{p} = o(l^{\beta_w - \beta_b})$, then $\mathbf{c} \to 1$, so that $\mathbf{e}^* = \frac{\frac{1}{2}\sigma_v^2 D \mathbb{J}_1(1) + \sigma_a^2}{\frac{1}{2}\sigma_v^2 D + \sigma_a^2} = 1$.

- If $\mathbf{p} \sim Cl^{\beta_w - \beta_b}$, then $\mathbf{c} \to \frac{\sigma_w^2 C \mathbf{e}^* + \sigma_b^2}{\sigma_w^2 C + \sigma_b^2}$, and we have an equation $\mathbf{e}^* = \frac{1}{\frac{1}{2}\sigma_v^2 D + \sigma_a^2}\left(\frac{1}{2}\sigma_v^2 D \mathbb{J}_1\left(\frac{\sigma_w^2 C \mathbf{e}^* + \sigma_b^2}{\sigma_w^2 C + \sigma_b^2}\right) + \sigma_a^2\right)$. By the monotonicity of $\mathbb{J}_1$, the RHS is at least $\frac{1}{\frac{1}{2}\sigma_v^2 D + \sigma_a^2}\left(\frac{1}{2}\sigma_v^2 D\left(\frac{\sigma_w^2 C \mathbf{e}^* + \sigma_b^2}{\sigma_w^2 C + \sigma_b^2}\right) + \sigma_a^2\right)$, which is a convex combination of $\mathbf{e}^*$ and 1. Since $\mathbf{e}^* \le 1$, the equality can hold only if $\mathbf{e}^* = 1$.

**Suppose $\mathcal{V}_r \le 1$ or $\mathcal{U}_r \le 1$.**

By Thm C.9, $\mathbf{p} = \Theta(1)$ and therefore $\boldsymbol{\gamma} = \Theta(1)$. Both converge to fixed points $\mathbf{p}^*$ and $\boldsymbol{\gamma}^*$ (possibly dependent on initial data $\mathbf{p}^{(0)}$ and $\boldsymbol{\gamma}^{(0)}$) because they are monotonically increasing sequences. Thus $\mathbf{e}^* = \boldsymbol{\gamma}^* / \mathbf{p}^*$.

Unwinding the proof of Thm C.9, we see that $\mathbf{p} - \underline{\mathbf{p}} = l^{-\mathcal{U}_r}$, so that $\mathbf{p}/(\mathbf{p} - \underline{\mathbf{p}}) = \Theta(l^{\mathcal{U}_r})$. Since the RHS of Eq. $(\star)$ cannot blow up, it must be the case that $(\epsilon - \underline{\epsilon}) = O(l^{-\mathcal{U}_r}) \implies \epsilon = O(l^{-\mathcal{U}_r + 1})$. If $(\epsilon - \underline{\epsilon})\mathbf{p}/(\mathbf{p} - \underline{\mathbf{p}}) = \mathcal{F} + o(1)$ for some constant $\mathcal{F}$, then the LHS becomes $\mathbf{e}^* + \mathcal{F}$ in the limit. Yet, unless $\boldsymbol{\gamma}^{(0)} = \mathbf{p}^{(0)}$, $\mathbf{e}^* < 1$. Therefore, $\mathcal{F} > 0$ (or else, like in the case of $\mathcal{V}_r \le 1$ or $\mathcal{U}_r \le 1$, $\mathbf{e}^* = 1$) whenever $\boldsymbol{\gamma}^{(0)} < \mathbf{p}^{(0)}$, and $\epsilon = \Theta(l^{-\mathcal{U}_r + 1})$.

$\square$

**Theorem C.11.** *Suppose $\phi = \mathrm{ReLU}$.*

- *If $\mathcal{V}_r > 1$, then $\boldsymbol{\chi}^{(0)}/\boldsymbol{\chi}^{(l)} = \Theta(1)$.*

  ○ *If $\mathcal{U}_r \in [0, 1)$*

$$\boldsymbol{\chi}_w^{(1)}/\boldsymbol{\chi}_w^{(l)} = \Theta(l^{\mathcal{U}_r - 1 + \beta_v}) \qquad\qquad \boldsymbol{\chi}_b^{(1)}/\boldsymbol{\chi}_b^{(l)} = \Theta(l^{\beta_v})$$
$$\boldsymbol{\chi}_v^{(1)}/\boldsymbol{\chi}_v^{(l)} = \Theta(l^{-\max(1 - \mathcal{U}_r - \beta_w, -\beta_b)}) \qquad\qquad \boldsymbol{\chi}_a^{(1)}/\boldsymbol{\chi}_a^{(l)} = \Theta(1).$$

  ○ *If $\mathcal{U}_r = 1$*
    ▪ *If $\beta_w \le \beta_b$*

$$\boldsymbol{\chi}_w^{(1)}/\boldsymbol{\chi}_w^{(l)} = \Theta(l^{\beta_v}/\log l) \qquad\qquad \boldsymbol{\chi}_b^{(1)}/\boldsymbol{\chi}_b^{(l)} = \Theta(l^{\beta_v})$$
$$\boldsymbol{\chi}_v^{(1)}/\boldsymbol{\chi}_v^{(l)} = \Theta(l^{\beta_w}/\log l) \qquad\qquad \boldsymbol{\chi}_a^{(1)}/\boldsymbol{\chi}_a^{(l)} = \Theta(1)$$

. *Otherwise*

$$\boldsymbol{\chi}_w^{(1)}/\boldsymbol{\chi}_w^{(l)} = \Theta(l^{\beta_v}/\log l) \qquad\qquad \boldsymbol{\chi}_b^{(1)}/\boldsymbol{\chi}_b^{(l)} = \Theta(l^{\beta_v})$$
$$\boldsymbol{\chi}_v^{(1)}/\boldsymbol{\chi}_v^{(l)} = \Theta(l^{\beta_b}) \qquad\qquad \boldsymbol{\chi}_a^{(1)}/\boldsymbol{\chi}_a^{(l)} = \Theta(1)$$

○ *If $\mathcal{U}_{\mathrm{r}} > 1$*

$$\boldsymbol{\chi}_w^{(1)}/\boldsymbol{\chi}_w^{(l)} = \Theta(l^{\beta_v}) \qquad\qquad \boldsymbol{\chi}_b^{(1)}/\boldsymbol{\chi}_b^{(l)} = \Theta(l^{\beta_v})$$
$$\boldsymbol{\chi}_v^{(1)}/\boldsymbol{\chi}_v^{(l)} = \Theta(l^{\min(\beta_w,\beta_b)}) \qquad\qquad \boldsymbol{\chi}_a^{(1)}/\boldsymbol{\chi}_a^{(l)} = \Theta(1)$$

• *If $\mathcal{V}_{\mathrm{r}} = 1$, then $\boldsymbol{\chi}^{(0)}/\boldsymbol{\chi}^{(l)} = \Theta(l^{\mathcal{W}_{\mathrm{r}}})$.*

○ *If $\mathcal{W}_{\mathrm{r}} \neq 1 - \mathcal{U}_{\mathrm{r}}$*

$$\boldsymbol{\chi}_w^{(1)}/\boldsymbol{\chi}_w^{(l)} = \Theta(l^{\mathcal{W}_{\mathrm{r}}+\beta_v-\max(\mathcal{W}_{\mathrm{r}},1-\mathcal{U}_{\mathrm{r}})}) \qquad\qquad \boldsymbol{\chi}_b^{(1)}/\boldsymbol{\chi}_b^{(l)} = \Theta(l^{\mathcal{W}_{\mathrm{r}}+\beta_v})$$
$$\boldsymbol{\chi}_v^{(1)}/\boldsymbol{\chi}_v^{(l)} = \Theta(l^{\mathcal{W}_{\mathrm{r}}-\max(\max(\mathcal{W}_{\mathrm{r}},1-\mathcal{U}_{\mathrm{r}})-\beta_w,-\beta_b)}) \qquad \boldsymbol{\chi}_a^{(1)}/\boldsymbol{\chi}_a^{(l)} = \Theta(l^{\mathcal{W}_{\mathrm{r}}})$$

○ *Otherwise*

. *If $\beta_b \geq \beta_w - \mathcal{W}_{\mathrm{r}}$*

$$\boldsymbol{\chi}_w^{(1)}/\boldsymbol{\chi}_w^{(l)} = \Theta(l^{\beta_v}/\log l) \qquad\qquad \boldsymbol{\chi}_b^{(1)}/\boldsymbol{\chi}_b^{(l)} = \Theta(l^{\mathcal{W}_{\mathrm{r}}+\beta_v})$$
$$\boldsymbol{\chi}_v^{(1)}/\boldsymbol{\chi}_v^{(l)} = \Theta(l^{\beta_w}/\log l) \qquad\qquad \boldsymbol{\chi}_a^{(1)}/\boldsymbol{\chi}_a^{(l)} = \Theta(l^{\mathcal{W}_{\mathrm{r}}})$$

. *Otherwise*

$$\boldsymbol{\chi}_w^{(1)}/\boldsymbol{\chi}_w^{(l)} = \Theta(l^{\beta_v}/\log l) \qquad\qquad \boldsymbol{\chi}_b^{(1)}/\boldsymbol{\chi}_b^{(l)} = \Theta(l^{\mathcal{W}_{\mathrm{r}}+\beta_v})$$
$$\boldsymbol{\chi}_v^{(1)}/\boldsymbol{\chi}_v^{(l)} = \Theta(l^{\mathcal{W}_{\mathrm{r}}+\beta_b}) \qquad\qquad \boldsymbol{\chi}_a^{(1)}/\boldsymbol{\chi}_a^{(l)} = \Theta(l^{\mathcal{W}_{\mathrm{r}}})$$

• *If $\mathcal{V}_{\mathrm{r}} < 1$, then $\boldsymbol{\chi}^{(0)}/\boldsymbol{\chi}^{(l)} = \exp(\frac{\mathcal{W}_{\mathrm{r}}}{1-\mathcal{V}_{\mathrm{r}}}l^{1-\mathcal{V}_{\mathrm{r}}} + \tilde{\Theta}(l^{\max(1-2\beta_v-2\beta_w,0)}))$.*

$$\boldsymbol{\chi}_w^{(1)}/\boldsymbol{\chi}_w^{(l)} = \Theta(1)$$
$$\boldsymbol{\chi}_b^{(1)}/\boldsymbol{\chi}_b^{(l)} = \exp(\frac{\mathcal{W}_{\mathrm{r}}}{1-\mathcal{V}_{\mathrm{r}}}l^{1-\mathcal{V}_{\mathrm{r}}} + \tilde{\Theta}(l^{\max(1-2\beta_v-2\beta_w,0)}))$$
$$\boldsymbol{\chi}_v^{(1)}/\boldsymbol{\chi}_v^{(l)} = \Theta(l^{\beta_w})$$
$$\boldsymbol{\chi}_a^{(1)}/\boldsymbol{\chi}_a^{(l)} = \exp(\frac{\mathcal{W}_{\mathrm{r}}}{1-\mathcal{V}_{\mathrm{r}}}l^{1-\mathcal{V}_{\mathrm{r}}} + \tilde{\Theta}(l^{\max(1-2\beta_v-2\beta_w,0)})).$$

*Proof.* Using Thm C.3 and the fact that $\mathrm{V}\phi(\mathbf{q}) = \frac{1}{2}\mathbf{q}, \mathrm{V}\dot{\phi}(\mathbf{q}) = \frac{1}{2}$, we get

$$\underline{\boldsymbol{\chi}} = (\frac{1}{2}\sigma_v^2\sigma_w^2 l^{-\mathcal{V}_{\mathrm{r}}} + 1)\boldsymbol{\chi}, \qquad\qquad \boldsymbol{\chi}_b = \frac{1}{2}\sigma_v^2 l^{-\beta_v}\boldsymbol{\chi},$$
$$\boldsymbol{\chi}_w = \frac{1}{2}\sigma_v^2 l^{-\beta_v}\boldsymbol{\chi}\underline{\mathbf{p}}, \qquad\qquad \boldsymbol{\chi}_v = \frac{1}{2}\boldsymbol{\chi}\mathbf{q}, \qquad\qquad \boldsymbol{\chi}_a = \boldsymbol{\chi},$$

so

$$\frac{\boldsymbol{\chi}_b^{(1)}}{\boldsymbol{\chi}_b^{(l)}} = \frac{\frac{1}{2}\sigma_v^2\boldsymbol{\chi}^{(1)}}{\frac{1}{2}\sigma_v^2 l^{-\beta_v}\boldsymbol{\chi}^{(l)}} = l^{\beta_v}\frac{\boldsymbol{\chi}^{(0)}}{\boldsymbol{\chi}^{(l)}}$$

$$\frac{\boldsymbol{\chi}_w^{(1)}}{\boldsymbol{\chi}_w^{(l)}} = \frac{\frac{1}{2}\sigma_v^2\boldsymbol{\chi}^{(1)}\mathbf{p}^{(0)}}{\frac{1}{2}\sigma_v^2 l^{-\beta_v}\boldsymbol{\chi}^{(l)}\mathbf{p}^{(l-1)}} = l^{\beta_v}\frac{\mathbf{p}^{(0)}}{\mathbf{p}^{(l-1)}}\frac{\boldsymbol{\chi}^{(0)}}{\boldsymbol{\chi}^{(l)}}$$

$$\frac{\boldsymbol{\chi}_v^{(1)}}{\boldsymbol{\chi}_v^{(l)}} = \frac{\frac{1}{2}\boldsymbol{\chi}^{(1)}\mathbf{q}^{(1)}}{\frac{1}{2}\boldsymbol{\chi}^{(l)}\mathbf{q}^{(l-1)}} = \frac{\mathbf{q}^{(0)}}{\mathbf{q}^{(l-1)}}\frac{\boldsymbol{\chi}^{(0)}}{\boldsymbol{\chi}^{(l)}}$$

$$\frac{\boldsymbol{\chi}_a^{(1)}}{\boldsymbol{\chi}_a^{(l)}} = \frac{\boldsymbol{\chi}^{(1)}}{\boldsymbol{\chi}^{(l)}}.$$

Suppose $\mathcal{V}_{\mathrm{r}} > 1$. Then $\frac{\boldsymbol{\chi}^{(0)}}{\boldsymbol{\chi}^{(l)}} = \Theta(1)$ by Lemma D.7. By Thm C.9:

- If $\mathcal{V}_r > 1$, then $\frac{\boldsymbol{\chi}^{(0)}}{\boldsymbol{\chi}^{(l)}} = \Theta(1)$ by Lemma D.7.

    ○ If $\mathcal{U}_r \in [0, 1)$
      ▪ $\mathbf{p} = \Theta(l^{1-\mathcal{U}_r})$ and $\mathbf{q} = \Theta(l^{\max(1-\mathcal{U}_r-\beta_w, -\beta_b)})$. So
      $$\boldsymbol{\chi}_w^{(1)}/\boldsymbol{\chi}_w^{(l)} = \Theta(l^{\mathcal{U}_r-1+\beta_v}) \qquad\qquad \boldsymbol{\chi}_b^{(1)}/\boldsymbol{\chi}_b^{(l)} = \Theta(l^{\beta_v})$$
      $$\boldsymbol{\chi}_v^{(1)}/\boldsymbol{\chi}_v^{(l)} = \Theta(l^{-\max(1-\mathcal{U}_r-\beta_w, -\beta_b)}) \qquad\qquad \boldsymbol{\chi}_a^{(1)}/\boldsymbol{\chi}_a^{(l)} = \Theta(1).$$

    ○ If $\mathcal{U}_r = 1$, then $\mathbf{p} = \Theta(\log l)$ and
      ▪ If $\beta_w \leq \beta_b$
        ◇ $\mathbf{q} = \Theta(l^{-\beta_w}\log l)$ so that
        $$\boldsymbol{\chi}_w^{(1)}/\boldsymbol{\chi}_w^{(l)} = \Theta(l^{\beta_v}/\log l) \qquad\qquad \boldsymbol{\chi}_b^{(1)}/\boldsymbol{\chi}_b^{(l)} = \Theta(l^{\beta_v})$$
        $$\boldsymbol{\chi}_v^{(1)}/\boldsymbol{\chi}_v^{(l)} = \Theta(l^{\beta_w}/\log l) \qquad\qquad \boldsymbol{\chi}_a^{(1)}/\boldsymbol{\chi}_a^{(l)} = \Theta(1)$$
      ▪ Otherwise
        ◇ $\mathbf{q} = \Theta(l^{-\beta_b})$ so that
        $$\boldsymbol{\chi}_w^{(1)}/\boldsymbol{\chi}_w^{(l)} = \Theta(l^{\beta_v}/\log l) \qquad\qquad \boldsymbol{\chi}_b^{(1)}/\boldsymbol{\chi}_b^{(l)} = \Theta(l^{\beta_v})$$
        $$\boldsymbol{\chi}_v^{(1)}/\boldsymbol{\chi}_v^{(l)} = \Theta(l^{\beta_b}) \qquad\qquad \boldsymbol{\chi}_a^{(1)}/\boldsymbol{\chi}_a^{(l)} = \Theta(1)$$

    ○ If $\mathcal{U}_r > 1$
      ▪ $\mathbf{p} = \Theta(1)$ and $\mathbf{q} = \Theta(l^{-\min(\beta_w, \beta_b)})$. So
      $$\boldsymbol{\chi}_w^{(1)}/\boldsymbol{\chi}_w^{(l)} = \Theta(l^{\beta_v}) \qquad\qquad \boldsymbol{\chi}_b^{(1)}/\boldsymbol{\chi}_b^{(l)} = \Theta(l^{\beta_v})$$
      $$\boldsymbol{\chi}_v^{(1)}/\boldsymbol{\chi}_v^{(l)} = \Theta(l^{\min(\beta_w, \beta_b)}) \qquad\qquad \boldsymbol{\chi}_a^{(1)}/\boldsymbol{\chi}_a^{(l)} = \Theta(1)$$

- If $\mathcal{V}_r = 1$, then by Lemma D.7 $\boldsymbol{\chi}^{(0)}/\boldsymbol{\chi}^{(l)} = \Theta(l^{\frac{1}{2}\sigma_v^2\sigma_w^2}) = \Theta(l^{\mathcal{W}_r})$.

    ○ If $\mathcal{W}_r \neq 1 - \mathcal{U}_r$
      ▪ $\mathbf{p} = \Theta(l^{\max(\mathcal{W}_r, 1-\mathcal{U}_r)})$, and $\mathbf{q} = \Theta(l^{\max(\max(\mathcal{W}_r, 1-\mathcal{U}_r)-\beta_w, -\beta_b)})$. So
      $$\boldsymbol{\chi}_w^{(1)}/\boldsymbol{\chi}_w^{(l)} = \Theta(l^{\mathcal{W}_r+\beta_v-\max(\mathcal{W}_r, 1-\mathcal{U}_r)}) \qquad \boldsymbol{\chi}_b^{(1)}/\boldsymbol{\chi}_b^{(l)} = \Theta(l^{\mathcal{W}_r+\beta_v})$$
      $$\boldsymbol{\chi}_v^{(1)}/\boldsymbol{\chi}_v^{(l)} = \Theta(l^{\mathcal{W}_r-\max(\max(\mathcal{W}_r, 1-\mathcal{U}_r)-\beta_w, -\beta_b)}) \quad \boldsymbol{\chi}_a^{(1)}/\boldsymbol{\chi}_a^{(l)} = \Theta(l^{\mathcal{W}_r})$$

    ○ Otherwise, $\mathbf{p} = \Theta(l^{\mathcal{W}_r}\log l)$, and
      ▪ If $\beta_b \geq \beta_w - \mathcal{W}_r$
        ◇ $\mathbf{q} = \Theta(l^{\mathcal{W}_r-\beta_w}\log l)$, so
        $$\boldsymbol{\chi}_w^{(1)}/\boldsymbol{\chi}_w^{(l)} = \Theta(l^{\mathcal{W}_r+\beta_v-\mathcal{W}_r}/\log l) = \Theta(l^{\beta_v}/\log l) \quad \boldsymbol{\chi}_b^{(1)}/\boldsymbol{\chi}_b^{(l)} = \Theta(l^{\mathcal{W}_r+\beta_v})$$
        $$\boldsymbol{\chi}_v^{(1)}/\boldsymbol{\chi}_v^{(l)} = \Theta(l^{\mathcal{W}_r+\beta_w-\mathcal{W}_r}/\log l) = \Theta(l^{\beta_v}/\log l) \quad \boldsymbol{\chi}_a^{(1)}/\boldsymbol{\chi}_a^{(l)} = \Theta(l^{\mathcal{W}_r})$$
      ▪ Otherwise
        ◇ $\mathbf{q} = \Theta(l^{-\beta_b})$, so
        $$\boldsymbol{\chi}_w^{(1)}/\boldsymbol{\chi}_w^{(l)} = \Theta(l^{\mathcal{W}_r+\beta_v-\mathcal{W}_r}/\log l) = \Theta(l^{\beta_v}/\log l) \quad \boldsymbol{\chi}_b^{(1)}/\boldsymbol{\chi}_b^{(l)} = \Theta(l^{\mathcal{W}_r+\beta_v})$$
        $$\boldsymbol{\chi}_v^{(1)}/\boldsymbol{\chi}_v^{(l)} = \Theta(l^{\mathcal{W}_r+\beta_b}) \qquad\qquad \boldsymbol{\chi}_a^{(1)}/\boldsymbol{\chi}_a^{(l)} = \Theta(l^{\mathcal{W}_r})$$

- If $\mathcal{V}_r < 1$, then by Lemma D.7, $\boldsymbol{\chi}^{(0)}/\boldsymbol{\chi}^{(l)} = \exp(\frac{\mathcal{W}_r}{1-\mathcal{V}_r}l^{1-\mathcal{V}_r} + R(l; \mathcal{W}_r, \mathcal{V}_r))$, where $R = \tilde{\Theta}(l^{\max(1-2\mathcal{V}_r, 0)})$ is as in Lemma D.7.

    ○ We have $\mathbf{p} = \exp(\frac{\mathcal{W}_r}{1-\mathcal{V}_r}l^{1-\mathcal{V}_r} + R(l; \mathcal{W}_r, \mathcal{V}_r) + O(1))$, where $R$ is the same as above, and $\mathbf{q} = \exp(\frac{\mathcal{W}_r}{1-\beta}l^{1-\mathcal{V}_r} + R(l; \mathcal{W}_r, \mathcal{V}_r) - \beta_w \log l + O(1))$. Thus

    $$\boldsymbol{\chi}_w^{(1)}/\boldsymbol{\chi}_w^{(l)} = \exp(O(1)) = \Theta(1)$$
    $$\boldsymbol{\chi}_b^{(1)}/\boldsymbol{\chi}_b^{(l)} = \exp(\frac{\mathcal{W}_r}{1-\mathcal{V}_r}l^{1-\mathcal{V}_r} + R + \beta_v \log l)$$
    $$\boldsymbol{\chi}_v^{(1)}/\boldsymbol{\chi}_v^{(l)} = \exp(\beta_w \log l + O(1)) = \Theta(l^{\beta_w})$$
    $$\boldsymbol{\chi}_a^{(1)}/\boldsymbol{\chi}_a^{(l)} = \exp(\frac{\mathcal{W}_r}{1-\mathcal{V}_r}l^{1-\mathcal{V}_r} + R).$$

$\square$

## D    LEMMAS

In this section, we present many lemmas used in the proofs of the main theorems. In all cases, the lemmas here have already appeared in some form in Yang and Schoenholz (2017), and for completeness, we either include them and their proofs here or improve upon and extend them, with the blessing of the authors.

**Lemma D.1.** *Asymptotically,*

$$
\mathbf{c} = \frac{\sigma_w^2 l^{-\beta_w}\underline{\gamma} + \sigma_b^2 l^{-\beta_b}}{\sigma_w^2 l^{-\beta_w}\underline{\mathbf{p}} + \sigma_b^2 l^{-\beta_b}}
$$

$$
= \begin{cases}
1 - \sigma_w^2\sigma_b^{-2}l^{\beta_b-\beta_w}\underline{\mathbf{p}}(1 - \underline{\gamma}/\underline{\mathbf{p}}) + \Theta(l^{2(\beta_b-\beta_w)}\underline{\mathbf{p}}^2(1 - \underline{\gamma}/\underline{\mathbf{p}})) \\
\qquad = 1 + O(l^{\beta_b-\beta_w}\underline{\mathbf{p}}) & \text{if } \underline{\mathbf{p}} = o(l^{\beta_w-\beta_b}) \\[2mm]
\underline{\gamma}/\underline{\mathbf{p}} + \sigma_b^2\sigma_w^{-2}l^{\beta_w-\beta_b}\underline{\mathbf{p}}^{-1}(\underline{\gamma}/\underline{\mathbf{p}} - 1) + \Theta(l^{2(\beta_w-\beta_b)}\underline{\mathbf{p}}^{-2}(\underline{\gamma}/\underline{\mathbf{p}} - 1)) \\
\qquad = \underline{\gamma}/\underline{\mathbf{p}} + O(l^{\beta_w-\beta_b}\underline{\mathbf{p}}^{-1}) & \text{if } \underline{\mathbf{p}} = \omega(l^{\beta_w-\beta_b}) \\[2mm]
\frac{\sigma_w^2 C\underline{\gamma}/\underline{\mathbf{p}}+\sigma_b^2}{\sigma_w^2 C+\sigma_b^2} + \mathcal{E}Rl^{\beta_b-\beta_w}(\underline{\gamma}/\underline{\mathbf{p}} - 1) + \Theta((Rl^{\beta_b-\beta_w})^2(\underline{\gamma}/\underline{\mathbf{p}} - 1)) \\
\qquad = \frac{\sigma_w^2 C\underline{\gamma}/\underline{\mathbf{p}}+\sigma_b^2}{\sigma_w^2 C+\sigma_b^2} + o(1) & \begin{array}{l}\text{if } \underline{\mathbf{p}} = Cl^{\beta_w-\beta_b} + R \text{ for} \\ \text{some constant } C \text{ and re-} \\ \text{mainder } R = o(l^{\beta_w-\beta_b}).\end{array}
\end{cases}
$$

*where* $\mathcal{E} = \frac{\sigma_w^2\sigma_b^2}{(\sigma_w^2 C+\sigma_b^2)^2}$.

*Proof.* If $\underline{\mathbf{p}} = o(l^{\beta_w-\beta_b})$, then

$$(\sigma_b^2 l^{-\beta_b} + \sigma_w^2 l^{-\beta_w}\underline{\mathbf{p}})^{-1} = \sigma_b^{-2}l^{\beta_b}(1 + \sigma_w^2\sigma_b^{-2}l^{\beta_b-\beta_w}\underline{\mathbf{p}})^{-1}$$

$$= \sigma_b^{-2}l^{\beta_b}(1 - \sigma_w^2\sigma_b^{-2}l^{\beta_b-\beta_w}\underline{\mathbf{p}} + (\sigma_w^2\sigma_b^{-2}l^{\beta_b-\beta_w}\underline{\mathbf{p}})^2 - \cdots)$$

$$\frac{\sigma_w^2 l^{-\beta_w}\underline{\gamma} + \sigma_b^2 l^{-\beta_b}}{\sigma_w^2 l^{-\beta_w}\underline{\mathbf{p}} + \sigma_b^2 l^{-\beta_b}} = (\sigma_w^2 l^{-\beta_w}\underline{\gamma} + \sigma_b^2 l^{-\beta_b})\sigma_b^{-2}l^{\beta_b}(1 - \sigma_w^2\sigma_b^{-2}l^{\beta_b-\beta_w}\underline{\mathbf{p}} + (\sigma_w^2\sigma_b^{-2}l^{\beta_b-\beta_w}\underline{\mathbf{p}})^2 - \cdots)$$

$$= (\sigma_w^2\sigma_b^{-2}l^{\beta_b-\beta_w}\underline{\gamma} + 1)(1 - \sigma_w^2\sigma_b^{-2}l^{\beta_b-\beta_w}\underline{\mathbf{p}} + (\sigma_w^2\sigma_b^{-2}l^{\beta_b-\beta_w}\underline{\mathbf{p}})^2 - \cdots)$$

$$= 1 - \sigma_w^2\sigma_b^{-2}l^{\beta_b-\beta_w}(\underline{\mathbf{p}} - \underline{\gamma}) + \Theta((l^{\beta_b-\beta_w})^2\underline{\mathbf{p}}(\underline{\mathbf{p}} - \underline{\gamma}))$$

$$= 1 + O(l^{\beta_b-\beta_w}\underline{\mathbf{p}}).$$

If $\underline{\mathbf{p}} = \omega(l^{\beta_w-\beta_b})$, then

$$(\sigma_w^2 l^{-\beta_w}\underline{\mathbf{p}} + \sigma_b^2 l^{-\beta_b})^{-1} = (\sigma_w^2 l^{-\beta_w}\underline{\mathbf{p}})^{-1}(1 + \sigma_b^2\sigma_w^{-2}l^{\beta_w-\beta_b}\underline{\mathbf{p}}^{-1})^{-1}$$

$$= (\sigma_w^2 l^{-\beta_w}\underline{\mathbf{p}})^{-1}(1 - \sigma_b^2\sigma_w^{-2}l^{\beta_w-\beta_b}\underline{\mathbf{p}}^{-1} + (\sigma_b^2\sigma_w^{-2}l^{\beta_w-\beta_b}\underline{\mathbf{p}}^{-1})^2 - \cdots)$$

$$\frac{\sigma_w^2 l^{-\beta_w}\underline{\gamma} + \sigma_b^2 l^{-\beta_b}}{\sigma_w^2 l^{-\beta_w}\underline{\mathbf{p}} + \sigma_b^2 l^{-\beta_b}} = (\underline{\gamma}/\underline{\mathbf{p}} + \sigma_b^2\sigma_w^{-2}l^{\beta_w-\beta_b}\underline{\mathbf{p}}^{-1})(1 - \sigma_b^2\sigma_w^{-2}l^{\beta_w-\beta_b}\underline{\mathbf{p}}^{-1} + (\sigma_b^2\sigma_w^{-2}l^{\beta_w-\beta_b}\underline{\mathbf{p}}^{-1})^2 - \cdots)$$

$$= \underline{\gamma}/\underline{\mathbf{p}} + \sigma_b^2\sigma_w^{-2}l^{\beta_w-\beta_b}\underline{\mathbf{p}}^{-2}(\underline{\gamma} - \underline{\mathbf{p}}) + \Theta(l^{2(\beta_w-\beta_b)}\underline{\mathbf{p}}^{-3}(\underline{\gamma} - \underline{\mathbf{p}}))$$

$$= \underline{\gamma}/\underline{\mathbf{p}} + O(l^{\beta_w-\beta_b}\underline{\mathbf{p}}^{-1}).$$

If $\underline{\mathbf{p}} \sim Cl^{\beta_w - \beta_b}$ for some $C$ and $R = \underline{\mathbf{p}} - Cl^{\beta_w - \beta_b} = o(l^{\beta_w - \beta_b})$, then

$$\sigma_w^2 l^{-\beta_w} \underline{\mathbf{p}} + \sigma_b^2 l^{-\beta_b} = l^{-\beta_w} \underline{\mathbf{p}} \left( \sigma_w^2 + \frac{\sigma_b^2 l^{-\beta_b}}{l^{-\beta_w} \underline{\mathbf{p}}} \right)$$

$$= l^{-\beta_w} \underline{\mathbf{p}} \left( \sigma_w^2 + \frac{\sigma_b^2}{C + Rl^{\beta_b - \beta_w}} \right)$$

$$= \frac{l^{-\beta_w} \underline{\mathbf{p}}}{C + Rl^{\beta_b - \beta_w}} \left( \sigma_w^2 (C + Rl^{\beta_b - \beta_w}) + \sigma_b^2 \right)$$

$$\sigma_w^2 l^{-\beta_w} \underline{\boldsymbol{\gamma}} + \sigma_b^2 l^{-\beta_b} = \frac{l^{-\beta_w} \underline{\mathbf{p}}}{C + Rl^{\beta_b - \beta_w}} \left( \sigma_w^2 \underline{\boldsymbol{\gamma}} / \underline{\mathbf{p}} (C + Rl^{\beta_b - \beta_w}) + \sigma_b^2 \right)$$

$$\frac{\sigma_w^2 l^{-\beta_w} \underline{\boldsymbol{\gamma}} + \sigma_b^2 l^{-\beta_b}}{\sigma_w^2 l^{-\beta_w} \underline{\mathbf{p}} + \sigma_b^2 l^{-\beta_b}} = \frac{\sigma_w^2 \underline{\boldsymbol{\gamma}} / \underline{\mathbf{p}} (C + Rl^{\beta_b - \beta_w}) + \sigma_b^2}{\sigma_w^2 (C + Rl^{\beta_b - \beta_w}) + \sigma_b^2}$$

$$= \left( \frac{\sigma_w^2 \underline{\boldsymbol{\gamma}} / \underline{\mathbf{p}} C + \sigma_b^2}{\sigma_w^2 C + \sigma_b^2} + \frac{\sigma_w^2 \underline{\boldsymbol{\gamma}} / \underline{\mathbf{p}} Rl^{\beta_b - \beta_w}}{\sigma_w^2 C + \sigma_b^2} \right) \left( 1 - \frac{\sigma_w^2 Rl^{\beta_b - \beta_w}}{\sigma_w^2 C + \sigma_b^2} + \cdots \right)$$

$$= \frac{\sigma_w^2 \underline{\boldsymbol{\gamma}} / \underline{\mathbf{p}} C + \sigma_b^2}{\sigma_w^2 C + \sigma_b^2} + Rl^{\beta_b - \beta_w} \left( \frac{\sigma_w^2 \underline{\boldsymbol{\gamma}} / \underline{\mathbf{p}}}{\sigma_w^2 C + \sigma_b^2} - \frac{\sigma_w^2 \underline{\boldsymbol{\gamma}} / \underline{\mathbf{p}} C + \sigma_b^2}{\sigma_w^2 C + \sigma_b^2} \frac{\sigma_w^2}{\sigma_w^2 C + \sigma_b^2} \right) + \cdots$$

$$= \frac{\sigma_w^2 \underline{\boldsymbol{\gamma}} / \underline{\mathbf{p}} C + \sigma_b^2}{\sigma_w^2 C + \sigma_b^2} + Rl^{\beta_b - \beta_w} \frac{\sigma_w^2}{\sigma_w^2 C + \sigma_b^2} \left( \underline{\boldsymbol{\gamma}} / \underline{\mathbf{p}} - \frac{\sigma_w^2 \underline{\boldsymbol{\gamma}} / \underline{\mathbf{p}} C + \sigma_b^2}{\sigma_w^2 C + \sigma_b^2} \right) + \cdots$$

$$= \frac{\sigma_w^2 \underline{\boldsymbol{\gamma}} / \underline{\mathbf{p}} C + \sigma_b^2}{\sigma_w^2 C + \sigma_b^2} + Rl^{\beta_b - \beta_w} \frac{\sigma_w^2}{\sigma_w^2 C + \sigma_b^2} \frac{\sigma_b^2 (\underline{\boldsymbol{\gamma}} / \underline{\mathbf{p}} - 1)}{\sigma_w^2 C + \sigma_b^2} + \Theta((Rl^{\beta_b - \beta_w})^2 (\underline{\boldsymbol{\gamma}} / \underline{\mathbf{p}} - 1))$$

$$= \frac{\sigma_w^2 \underline{\boldsymbol{\gamma}} / \underline{\mathbf{p}} C + \sigma_b^2}{\sigma_w^2 C + \sigma_b^2} + o(1)$$

$\square$

For any function $f$ that is $(k+1)$-times differentiable in a neighborhood of $0$, we have the asymptotic expansion

$$f(z) = \sum_{n=0}^{k} \frac{d^n f}{dz^n}(0) \frac{z^n}{n!} + O(z^{k+1}), \text{ as } z \to 0.$$

Since

$$\frac{d^n}{d(1/\mathbf{q})^n} \mathbf{q}^{1/2} \mathrm{V}\phi(\mathbf{q}) \bigg|_{\mathbf{q} \to \infty} = \frac{(-1)^n}{2^n \sqrt{2\pi}} \int_{-\infty}^{\infty} \phi^2(z) z^{2n} \, \mathrm{d}z$$

whenever the RHS is integrable, we have

**Lemma D.2.** *Suppose $\phi^2(z) z^{2n}$ is integrable over $z \in \mathbb{R}$ for all $0 \le n \le N+1$. Then $\mathrm{V}\phi(\mathbf{q}) = \mathbf{q}^{-1/2} (\sum_{n=0}^{N} C_n \mathbf{q}^{-n} + O(\mathbf{q}^{-N-1}))$ as $\mathbf{q} \to \infty$, where*

$$C_n := \frac{(-1)^n}{2^n n! \sqrt{2\pi}} \int_{-\infty}^{\infty} \phi^2(z) z^{2n} \, \mathrm{d}z.$$

Note that $\mathrm{sech}^d(z) = \Theta(e^{-d|z|})$ for $z \to \infty$ as long as $d > 0$, so that $C_n$ from the above result converges when $\phi = \mathrm{sech}^d$. Therefore

**Lemma D.3.** *Let $d > 0$. We have $\mathrm{V} \, \mathrm{sech}^d(\mathbf{q}) \simeq \mathbf{q}^{-1/2} \sum_{n \ge 0} C_n \mathbf{q}^{-n}$, where*

$$C_n := \frac{(-1)^n}{2^n n! \sqrt{2\pi}} \int_{-\infty}^{\infty} \mathrm{sech}^{2d}(z) z^{2n} \, \mathrm{d}z.$$

As corollaries, we obtain the following asymptotics.

**Lemma D.4.** $\mathrm{V} \dot{\tanh}(\mathbf{q}) = \frac{2}{3} \sqrt{\frac{2}{\pi}} \mathbf{q}^{-1/2} + \Theta(\mathbf{q}^{-3/2})$ *as* $\mathbf{q} \to \infty$.

*Proof.* Use Lemma D.3 along with the fact that $\dot{\tanh}(z) = \text{sech}^2(z)$ and $\int \text{sech}^4 z \, dz = \frac{2}{3} \tanh z + \frac{1}{2} \text{sech}^2 z \tanh z$. $\qquad\square$

**Lemma D.5.** $1 - V \tanh(\mathbf{q}) = \sqrt{\frac{2}{\pi}} \mathbf{q}^{-1/2} + \Theta(\mathbf{q}^{-3/2})$ *as* $\mathbf{q} \to \infty$.

*Proof.* Use Lemma D.3 along with the fact that $1 - \tanh^2(z) = \text{sech}^2(z)$ and $\int \text{sech}^2 z \, dz = \tanh z$. $\qquad\square$

**Lemma D.6.** *Let* $d \in \mathbb{R}$ *and* $1 < M < N$ *with* $N - M \in \mathbb{Z}^{\geq 0}$. *Set* $\Sigma(M, N, d) := \sum_{a=M}^{N} a^d$. *If we fix* $M$ *and let* $N \to \infty$,

$$\Sigma(M, N, d) = \begin{cases} \Theta(1) & \text{if } d < -1 \\ \log N + O(1) & \text{if } d = -1 \\ \frac{N^{d+1}}{d+1} + O(1) & \text{if } -1 < d < 0 \\ N - M + 1 & \text{if } d = 0 \\ \frac{1}{d+1} N^{d+1} + \frac{1}{2} N^d + O(N^{\max(0, d-1)}) & \text{if } d > 0 \end{cases}$$

*Proof.* Euler-MacLaurin formula. $\qquad\square$

**Lemma D.7.** *Suppose* $\epsilon^{(l)}$ *satisfies the recurrence*

$$\epsilon^{(l)} = \epsilon^{(l-1)}\left(1 + \frac{\delta}{l^\beta}\right).$$

*for some nonzero constant* $\delta \in \mathbb{R}$ *independent of* $l$.

- *If* $\beta > 1$, *then* $\epsilon^{(l)} = \Theta(1)$.

- *If* $\beta = 1$, *then* $\epsilon^{(l)} = \Theta(l^\delta)$.

- *If* $0 < \beta < 1$, *then* $\epsilon^{(l)} = \exp(\frac{\delta}{1-\beta} l^{1-\beta} + \tilde{\Theta}(l^{\max(0, 1-2\beta)}))$. *In particular,* $\epsilon^{(l)} = \exp(\frac{\delta}{1-\beta} l^{1-\beta} + R)$, *where remainder* $R$ *is*

$$R = \begin{cases} \Theta(l^{1-2\beta}) & \text{if } \beta < 1/2 \\ \Theta(\log l) & \text{if } \beta = 1/2 \\ \Theta(1) & \text{if } \beta > 1/2 \end{cases}$$

*Proof.* We have

$$\log \epsilon^{(l)} = \log \epsilon^{(l-1)} + \log(1 + \delta/l^\beta)$$
$$= \log \epsilon^{(l-1)} + \delta/l^\beta + \Theta(\delta^2/l^{2\beta})$$

for large $l$. If $\beta > 1$, then $\sum_l l^{-\beta}$ converges, and

$$\log \epsilon^{(l)} = \log \epsilon^{(0)} - \Theta(1)$$
$$\epsilon^{(l)} = \Theta(1).$$

If $\beta = 1$, then

$$\log \epsilon^{(l)} = \log \epsilon^{(0)} + \delta \log l + \Theta(1)$$
$$\epsilon^{(l)} = \Theta(l^\delta).$$

If $\beta < 1$, then

$$\log \epsilon^{(l)} = \log \epsilon^{(0)} + \frac{\delta}{1-\beta} l^{1-\beta} + R$$

for a remainder $R$ that is

$$\begin{cases} \Theta(l^{1-2\beta}) & \text{if } \beta < 1/2 \\ \Theta(\log l) & \text{if } \beta = 1/2 \\ \Theta(1) & \text{if } \beta > 1/2 \end{cases}$$

Exponentiating gives the desired result. $\qquad\square$

**Lemma D.8.** *Suppose $\epsilon^{(l)} = Cl^{-\alpha} + \epsilon^{(l-1)}(1 + \delta/l^\beta)$ for $\alpha \in \mathbb{R}$, $C \neq 0$, and $\delta \neq 0$. Then*

- *If $\beta > 1$, then*
  - $\epsilon^{(l)} = \Theta(l^{1-\alpha})$ *if $\alpha \in [0,1)$;*
  - $\epsilon^{(l)} = \Theta(\log l)$ *if $\alpha = 1$;*
  - $\epsilon^{(l)} = \Theta(1)$ *if $\alpha > 1$.*

- *If $\beta = 1$, then*
  - $\epsilon^{(l)} = \Theta(l^{\max(\delta, 1-\alpha)})$ *if $1 - \delta \neq \alpha$.*
  - $\epsilon^{(l)} = \Theta(l^\delta \log l)$ *if $1 - \delta = \alpha$.*

- *If $\beta < 1$, then*
  - $\epsilon^{(l)} = \exp(\frac{\delta}{1-\beta}l^{1-\beta} + \tilde{\Theta}(l^{\max(0,1-2\beta)}))$. *In particular, $\epsilon^{(l)} = \exp(\frac{\delta}{1-\beta}l^{1-\beta} + R(l; \delta, \beta) + O(1))$, where $R$ is the same $R$ as in Lemma D.7, depending on only $l$, $\delta$, and $\beta$ but not on $\alpha$ or $C$, with*

$$R = \begin{cases} \Theta(l^{1-2\beta}) & \textit{if } \beta < 1/2 \\ \Theta(\log l) & \textit{if } \beta = 1/2 \\ \Theta(1) & \textit{if } \beta > 1/2. \end{cases}$$

*Furthermore, for $\beta = -\delta = 1$: $\epsilon^{(l)} \sim l^{-1}$ if $\alpha > 2$, $\epsilon^{(l)} \sim l^{1-\alpha}$ if $\alpha < 2$, and $\epsilon^{(l)} \sim l^\delta \log l$ if $\alpha = 2$.*

*Proof.* We can unwind the recurrence to get

$$\epsilon^{(l)} = C \sum_{m=1}^{l} m^{-\alpha} \prod_{n=m+1}^{l} \left(1 + \frac{\delta}{n^\beta}\right) + \epsilon^{(0)} \prod_{n=1}^{l}\left(1 + \frac{\delta}{n^\beta}\right)$$

Suppose $\beta > 1$. By Lemma D.7, we get

$$\epsilon^{(l)} = \Theta(1) \sum_{m=1}^{l} m^{-\alpha} + \epsilon^{(0)}\Theta(1)$$

$$= \begin{cases} \Theta(l^{1-\alpha}) & \text{if } \alpha \in [0,1) \\ \Theta(\log l) & \text{if } \alpha = 1 \\ \Theta(1) & \text{if } \alpha > 1. \end{cases}$$

Now suppose $\beta = 1$. By Lemma D.7, we get

$$\epsilon^{(l)} = \sum_{m=1}^{l} m^{-\alpha}\Theta(m^{-\delta}l^\delta) + \epsilon^{(0)}\Theta(l^\delta)$$

where the constants hidden inside the $\Theta$ are the same in every term of the sum. If $\alpha > 1 - \delta$, then $m^{-\delta-\alpha} = o(m^{-1})$, so that $\sum_{m=1}^{l} m^{-\delta-\alpha} = \Theta(1)$, and

$$\epsilon^{(l)} = \Theta(l^\delta) + \epsilon^{(0)}\Theta(l^\delta)$$
$$= \Theta(l^\delta).$$

On the other hand, if $\alpha < 1 - \delta$, then $\sum_{m=1}^{l} m^{-\delta-\alpha} = \Theta(l^{1-\delta-\alpha})$. So

$$\epsilon^{(l)} = \Theta(l^{1-\alpha}) + \epsilon^{(0)}\Theta(l^\delta)$$
$$= \Theta(l^{1-\alpha}).$$

x If $\alpha = 1 - \delta$, then $\sum_{m=1}^{l} m^{-\delta-\alpha} = \Theta(\log l)$. So

$$\epsilon^{(l)} = \Theta(l^\delta \log l) + \epsilon^{(0)}\Theta(l^\delta)$$
$$= \Theta(l^\delta \log l).$$

Finally, if $\beta \in (0, 1)$, then by Lemma D.7,

$$\epsilon^{(l)} = Ce^{\frac{\delta}{1-\beta}l^{1-\beta}+R} \sum_{m=1}^{l} m^{-\alpha} e^{\frac{-\delta}{1-\beta}m^{1-\beta}+\tilde{\Theta}(l^{\max(0,1-2\beta)})} + e^{\frac{\delta}{1-\beta}l^{1-\beta}+R},$$

where $R$ is the remainder as in Lemma D.7. The sum $\sum_{m=1}^{l} m^{-\alpha} e^{\frac{-\delta}{1-\beta}m^{1-\beta}+\tilde{\Theta}(l^{\max(0,1-2\beta)})}$ can be upper and lower bounded by integrals of the form $\int x^{-\alpha} e^{-\mu x^{1-\beta}} \, dx$ for appropriate $\mu$s, which are finite (they are bounded by values of incomplete Gamma functions). Thus $\epsilon^{(l)} = \Theta(1)e^{\frac{\delta}{1-\beta}l^{1-\beta}+R}$ where

$$R' = R + O(1) = \begin{cases} \Theta(l^{1-2\beta}) & \text{if } \beta < 1/2 \\ \Theta(\log l) & \text{if } \beta = 1/2 \\ \Theta(1) & \text{if } \beta > 1/2. \end{cases}$$

A fortiori, $\epsilon^{(l)} = e^{\frac{\delta}{1-\beta}l^{1-\beta}+\tilde{\Theta}(l^{\max(0,1-2\beta)})}$.

For our "furthermore" claim: the case of $\delta = -1$ telescopes, so that the upper and lower constants hidden in $\Theta$ can both be taken to be 1. $\qquad \square$

