# OpenReview forum: "Deep Mean Field Theory: Layerwise Variance and Width Variation as Methods to Control Gradient Explosion"
_ICLR.cc/2018/Conference — Invite to Workshop Track_

### Official Review · AnonReviewer1 · 2017-11-28
**Nice addition to the mean-field-theory subfield**

**Rating:** 7
**Confidence:** 3

**Review:**

This paper further develops the research program using mean field theory to predict generalization performance of deep neural networks. As with all recent mean-field papers, the main query here is to what extent the assumptions (Axioms 1+2, which basically define the asymptotic parameters of interest to be the quantities defined in Sec. 2.; and also the fully connected residual structure of the network) apply in practice. This is answered using the same empirical standard as in [Yang and Schoenholz, Schoenholz et al.], i.e. showing that the dynamics of initialization predict generalization behavior on MNIST according to theory.

As with the earlier papers in this recent program, the paper is notation-heavy but generally written well, though there is some overreliance on the readers' knowledge of previous work, for instance in presenting the evidence as above. Try as I might, I cannot find a detailed explanation of the color scale for the important Fig. 4. A small notation issue: the current Hebrew letter for the gradient quantity does not go with the other Greek letters and is typographically poor choice because of underlining, etc.). Also, several of the citations should be fixed to reflect peer-reviewed publication of Arxiv papers. I was not able to review all the proofs, but what I checked was sound. Finally, the techniques of WV and VV would be more applicable if it were not for the very tenuous relationship between gradient explosion and performance, which should be mentioned more than the one time it appears in the paper.

---

> ### Author Response · Authors · 2018-01-05
> **Thank you for taking the time to read and review our paper.**
>
> We respond to your comments as follows.
>
> > As with the earlier papers in this recent program, the paper is notation-heavy but generally written well, though there is some overreliance on the readers' knowledge of previous work, for instance in presenting the evidence as above.
>
> Thank you for your kind review. We agree that this overreliance has lead to poor presentation of our results. We have significantly rewritten our main text, devoting much space to summarizing the previous work and context, while toning down the heaviness of notation and technicality in favor of more intuitive discussion. See the changelog for a full list of changesl
>
> > Try as I might, I cannot find a detailed explanation of the color scale for the important Fig. 4.
> Thank you for pointing this out. We have added color bars to our heatmaps.
>
> > A small notation issue: the current Hebrew letter for the gradient quantity does not go with the other Greek letters and is typographically poor choice because of underlining, etc.).
> We have changed the Hebrew daleth to the Greek letter Chi, and bolded all mean field quantities to make them more readable. We have also compiled a symbol glossary to ameliorate the notation heaviness of our paper.
>
> > Also, several of the citations should be fixed to reflect peer-reviewed publication of Arxiv papers.
> Thank you for pointing out the error. We have updated the citations accordingly.
>
> > I was not able to review all the proofs, but what I checked was sound.
>
> > Finally, the techniques of WV and VV would be more applicable if it were not for the very tenuous relationship between gradient explosion and performance, which should be mentioned more than the one time it appears in the paper.
>
> It is true that, as Yang and Schoenholz observed in their NIPS 2017 paper, ReLU resnets are not bottlenecked by trainability but rather by (metric) expressivity. This is what we find in the zig phase of ReLU resnet VV, where metric expressivity predicts performance. However, VV does indeed decrease the activation explosion of ReLU resnets to prevent forward computation from overflowing.
>
> In the updated version of our paper, we have included our experiments on applying VV to tanh resnets, and there variance decay does improve performance by reducing gradient explosion. This is apparent in our figure 3 (in the new version), which shows that the optimal variance decay is larger for larger depth L. Again, this is expected based on Yang and Schoenholz's observation that tanh resnets are bottlenecked by trainability when variances are too large.
>
> Let us know if you are satisfied with our responses.

---

### Official Review · AnonReviewer3 · 2017-12-01
**Difficult to follow for someone not familiar with all the terminologies used in deep nets.**

**Rating:** 5
**Confidence:** 1

**Review:**

The authors study mean field theory for deep neural nets.

To the best of my knowledge we do not have a good understanding of mean field theory for neural networks and this paper  and some references therein are starting to address some of it.

However, my concern about the paper is in readability. I am very familiar with the literature on mean field theory but less so on deep nets. I found it difficult to follow many parts because the authors assume that the reader will have the knowledge of all the terminology in the paper, which there is a lot of.

---

> ### Author Response · Authors · 2018-01-05
> **Thank you for your review.**
>
> We have revamped the presentation of the paper, improving its presentation and addressing your concerns in readability. We hope you can give it another read.

---

### Official Review · AnonReviewer4 · 2017-12-13
**Missing literature, figures lack clarity, but conceptually interesting**

**Rating:** 5
**Confidence:** 3

**Review:**

Mean field theory is an approach to analysing complex systems where correlations between highly dependent random variables are ignored, thus making the problem analytically tractable. It is hoped that analytical insights gained in this idealised setting might translate back to the original (and far messier) problem. The authors use a mean field theory approach to study how varying certain network hyperparameters with depth can effect gradient and activation statistics. A correlation between the behaviour of these statistics and training performance on MNIST is noted.

As someone asked to conduct an 'emergency' review of this paper, I would have greatly appreciated the authors making more of an effort to present their results clearly. Some general comments in this regard:

Clarity issues:
- the authors appear to have ignored the ICLR style guidelines
- the references are all written in green, making them difficult to read
- figures are either missing color maps or make poor choice of colors
- the figure captions are difficult to understand in isolation from the main text
- the authors themselves appear to muddle their 'zigs' and 'zags' (first line of discussion)

Now to get to the actual content of the paper. The authors do not properly place their work in context. Mean field theory has been studied in the context of neural networks at least since the 80's. Entire books have been written on the statistical mechanics of neural networks. It seems wrong that the authors only cite papers on this matter going back to 2016.

With that said, the main thrust of the paper is very interesting. The authors derive recurrence relations for mean activations and gradients. They show how scaling layer width and initialisation variance with depth can better control the propagation of these means. The results of their calculations appear to match their random network simulations, and this part of the work seems strong.

What is not clear is what effect we should expect these quantities to have on learning? The authors claim there is a tradeoff between expressivity and exploding gradients. This seems quite speculative since it is not clear to me what effect either of these things will have on training. For one, how expressive does a model need to be to correctly classify MNIST? And are exploding gradients necessarily a bad thing? Provided they do not reach infinity, can we not just choose a smaller learning rate?

I'm open to reevaluating the review if the issues of clarity and missing literature review are fixed.

---

> ### Author Response · Authors · 2018-01-05
> **Thank you for your review.**
>
> We appreciate you answering the emergency call to review our paper.
> Our responses are as follows.
>
> > Clarity issues:
> > - the authors appear to have ignored the ICLR style guidelines
> In the new version, we have done the following:
> Abstract merged into 1 paragraph.
> Changed table title to be lower case except first word and pronoun.
> We have put parentheses around tail citations.
> Please let us know if you found more violations of the style guideline.
>
> > - the references are all written in green, making them difficult to read
> We thought that they actually improve readability, but based on your suggestion we have turned off colored links.
>
> > - figures are either missing color maps or make poor choice of colors
> Thank you for pointing this out. We have added color bars and improved color choices, especially in the heatmaps and their contour overlays.
>
> > - the figure captions are difficult to understand in isolation from the main text
> In response to your feedback, we have made figure captions much more self-contained.
>
> > - the authors themselves appear to muddle their 'zigs' and 'zags' (first line of discussion)
> Thanks for pointing out this error. It has been fixed.
>
> > Now to get to the actual content of the paper. The authors do not properly place their work in context. Mean field theory has been studied in the context of neural networks at least since the 80's. Entire books have been written on the statistical mechanics of neural networks. It seems wrong that the authors only cite papers on this matter going back to 2016.
>
> We apologize for this omission. In the new version, a significant chunk of the introduction is used for surveying previous works on mean field theory of neural networks.

---

> > ### Author Response · Authors · 2018-01-06
> > **Response (cont)**
> >
> > > The authors claim there is a tradeoff between expressivity and exploding gradients. This seems quite speculative since it is not clear to me what effect either of these things will have on training. For one, how expressive does a model need to be to correctly classify MNIST?
> >
> > We want to first make the following clarification: We are only claiming there is an effect on relative performance, i.e. we can say that one initialization achieves weakly better results (in particular, weakly better learning curves) than another initialization. We are NOT saying that that by initializing a certain way, you can solve MNIST or imagenet. We admit that we have not been sufficiently clear in the paper, and have stressed this point from the get-go in the updated version.
> >
> > Gradient explosion/vanishing is one of the most famous obstacles to training deep neural networks; see Bengio et al. (1994) and Pascanu et al. (2013), for example. The former noted that much of the difficulty of training RNNs arise from such gradient problems. In fact, in that paper already, the notion of expressivity vs trainability has arised: it is easy for an RNN to suffer from gradient explosion/vanishing problems when it tries to learn long time dependencies (striving to be expressive).
> >
> > The form of the claim specific to our case originates in Yang and Schoenholz (2017). There the authors made the observation that the optimal initialization scheme for tanh resnets makes an optimal tradeoff between expressivity and trainability: if the initialization variances are too big, then the random network will suffer from gradient explosion with high probability; if they are too small, then the random network will be approximately constant (i.e. has low metric expressivity) with high probability. Metric expressivity of a random network is the expectation of ||f(x) - f(x’)||^2, where f is the random net and x and x’ are two different input vectors. It measures how much the network expands the input space, on average. Intuitively, a larger metric expressivity means that it is easier to tell apart two vectors from their neural network embeddings via a linear separator.
> > This claim is strongly corroborated by their experiments with tanh and ReLU resnets.
> >
> > In our paper, we see this tradeoff determining the outcome of experiments in all but one case (ReLU resnet in the zag phase). We discuss this tradeoff at length in our revised paper, but we provide a summary below in case the reviewer does not have time to look at it.
> >
> > We confirm this behavior in tanh resnets when decaying their initialization variances with depth: When there is no decay, gradient explosion bottlenecks the test set accuracy after training; when we impose strong decay, gradient dynamics is mollified but then metric expressivity (essentially the average distance between the images of two different input vectors), being strongly constrained, caps the performance.
> > Indeed, we can predict test set accuracy by level curves of the magnitude of gradient explosion in the region of small variance decay, while we can do the same with level curves of metric expressivity when in the region of large decay. The performance peaks at the intersection of these two regions. Please see our experimental section in VV for more details.
> >
> > With ReLU resnets, there are two phases of behavior when we apply VV. In one (the zig phase), we start applying variance decay to some parameters (w and b). We see what is very similar to Yang and Schoenholz's observation, that decaying the variance prevents training failure from numerical overflow, but decaying it further reduces test time accuracy by reducing metric expressivity. This is consistent with the tradeoff: Our ReLU resnets in this zig phase have fairly tame gradient explosion (polynomial with low degree) while the metric expressivity is growing superpolynomially with depth, so the latter naturally dominates the effect on performance.
> >
> > In the other (zag) phase, which continues from the zig phase, we start decaying variances of other parameters. Here we observe a seeming counterexample to this tradeoff: weight gradient explosion worsens and expressivity decreases but the test set accuracy increases! In this phase, both metric expressivity and gradient explosion have polynomial dynamics with low degrees. So plausibly, a new factor begins to dominate the effect on performance that we do not know about yet.

---

> > > ### Author Response · Authors · 2018-01-06
> > > **Response (cont. 2)**
> > >
> > >
> > > > And are exploding gradients necessarily a bad thing?
> > >
> > > This is a great question. “Conventional wisdom” (starting from Bengio et al. (1994)) posits that they are always bad for training a deep net, and Pascanu et al. hypothesized that the reason is the ill-conditioning of the Hessian.
> > >
> > >
> > > In the updated version of our paper, we show this hypothesis is true if we replace “Hessian” with “Fisher information matrix” (which is the Hessian for KL divergence). See our new section 2 for details. Thus we do expect concrete optimization obstacles when there is gradient explosion/vanishing.
> > >
> > >
> > > In the context of random networks, this is supported experimentally by recent works by Schoenholz et al. (2017) and Yang and Schoenholz (2017), where optimal initializations are those that avoid gradient explosion (without losing too much expressivity). This is also supported by our new experiments on applying VV to tanh resnets, where imposing stronger variance decay improves performance (until the point where metric expressivity drops too much).
> > >
> > >
> > > But our ReLU experiments also show that mysteriously, in the zag regime of VV for ReLU resnets, larger weight gradients correlate with better performance, and we do not know how to explain it any other way. Thus your question reflects exactly one point raised by our work: are there in fact scenarios where greater gradient explosion can actually cause better performance? We hope to answer this in the future.
> > >
> > >
> > > > Provided they do not reach infinity, can we not just choose a smaller learning rate?
> > >
> > >
> > > In fact a “smaller learning rate” was essentially what Pascanu et al. proposed --- gradient clipping --- and remains one of the most popular ways to deal with gradient explosion when they occur. However, as discussed in our new section 2, gradient explosion causes optimization difficulties in the way of ill-conditioned Fisher information. In the case when we are actually minimizing the KL divergence so that Fisher information is in fact its Hessian, this ill-conditioning presents an obstruction to first order optimization methods, regardless of learning rate. Please see our text for details. We want to stress that gradient explosion is not simply a matter of gradient magnitude too big, but rather an issue where the first few layers of a deep network gets "more error signals" in the form of gradients than the last few. Multiplying every gradient term by  the same learning rate does not change this circumstance. This "information propagation" perspective is in fact the theme of Schoenholz et al. (2017).
> > >
> > > We do agree however that more research is needed to decipher the cross effect of learning rate and initialization. Work is currently underway.

---

### Author Response · Authors · 2018-01-05
**Changelog**

We have updated our paper as follows:
1.	We added a new section that elucidates the gradient explosion/vanishing problem from an information geometry perspective. We reason that this problem manifests in the exponential ill-conditioning of the Fisher information matrix, so that (stochastic) gradient descent approximates the natural gradient poorly.
2.	We added experiments on applying VV to tanh resnets. We find that variance decay improves performance of tanh resnets. In particular, the optimal decay cannot be too small nor too large, but rather must balance trainability and expressivity.
3.	We added a background section summarizing the recent line of work that we are building on and discuss how our work relates to them.
4.	We added a section overviewing our techniques and main results in intuitive terms. In particular, we devote a significant chunk to discussing the trainability vs expressivity tradeoff.
5.	We devoted significant space in the introduction to discuss prior works in mean field theory and recent trends.
6.	We swapped out the Hebrew letters for better alternatives; for example, Hebrew daleth is now chi. We also bolded all mean field quantities to improve readability.
7.	We added a notation glossary to improve readability.
8.	We improved colors and presentations of the plots, especially the heatmaps and overlaid contours. We also added color bars.
9.	We moved the detailed discussion on the VV dynamics in the original manuscript to the appendix, and only sketch the key points in enough detail in the main text for the experiments to make sense to the reader.
10.	We moved discussion of mean field assumption to the appendix, as they might be confusing to the first time reader.
11.	Similarly we moved definition of the integral operators V and W, along with the table of dynamical equations we derive in this paper, to the appendix, to decrease notation baggage. Most of the main text can be understood without examining these details.
12.	We rewrote figure captions to be self-contained.
13.	We fixed various ICLR style guideline issues.
14.	We turned off colored links.
15.	We fixed various typos and grammatical mistakes.

---

### Decision · Program_Chairs · 2018-01-29
**ICLR 2018 Conference Acceptance Decision**

**Decision:**

Invite to Workshop Track

**Comment:**

All the reviewers agree that this is an interesting paper but have concerns about readability and presentation. There is also concern that many results are speculative and not concretely tested. I recommend the authors to carefully investigate their claims with stronger experiments and submit it to another venue. I recommend presenting at ICLR workshop to obtain further feedback.